# Adam Kraft's Moving Sandstones

Larry Silver

Department of Art History, University of Pennsylvania, Philadelphia, PA 19104, USA; lasilver@sas.upenn.edu

**Abstract:** Adam Kraft, Albrecht Dürer's contemporary in Nuremberg, worked in the material of sandstone to provide a comparable experience in carved relief about the Passion of Christ. Both artists began their work in Nuremberg around the same time, 1490, although the older Kraft actually predeceased Dürer by two full decades (1508/1528). But both Nuremberg artists shared a religious sentiment of late-medieval art as having a goal to evoke pious emotions through vivid, multi-figured narrative re-enactments. Kraft's *Stations of the Cross* series simulates an imaginary pilgrimage in Jerusalem itself. Through their visual process, both Kraft and Dürer moved pious empathy in their—literally—moving viewers of Passion sequences.

**Keywords:** Adam Kraft; Nuremberg; sandstone; relief sculpture; Passion of Christ; virtual pilgrimage

The challenge of this special issue of *Arts* is to consider moving images. This paper will consider specific examples of Christian imagery as "moving" in two senses, both as physical motion and as arousal of pious emotion. These two aspects of viewer responses were elicited by the torment-laden Christian narrative of the Passion of Jesus, specifically by the unfolding process of his torments, death, and burial. In particular, following the precedent for sequential experience of the Passion, established by German printmakers in the late fifteenth century, the Nuremberg sculptor Adam Kraft (active 1490–1508; Figure 1) took up a similar challenge: to produce affective Passion imagery in sculpted sandstone reliefs.

Kraft provided both kinds of "moving imagery" in his Passion representation. His first reliefs wrapped the Passion scenes around a church apse at Nuremberg's parish church of St Sebald's. This arrangement obliged a pious view to move laterally in order to follow the narrative in sequence. But Kraft also arranged his scenes so that they unfolded "against the grain," from right to left. Thus, they created a subconscious resistance to regular left-to-right viewing habits, reinforcing the emotional content of these scenes as obstacles and torments to normal progression.

Later, Kraft replicated Jesus's own Way of the Cross or procession to Calvary in a sequence of separate reliefs erected in local imitation from the Nuremberg city walls to the civic cemetery of St. John's. In following those events all the way to a simulated Tomb of the Holy Sepulchre, a celebrated rite of Jerusalem pilgrims on site, the 16th-century German Christian could experience a vicarious personal Passion. The Nuremberg procession by Kraft thus incorporated both physical movement along a sacred route as well as a series of emotional responses to the depicted burden of the cross amid the tortures imposed on Jesus by his mocking tormentors.

Adam Kraft worked throughout his career under the rules of art-making in Nuremberg. Uniquely among German sixteenth-century cities, Nuremberg had no formal guild structures (Brandl 1986). Local painters and sculptors, however, enjoyed another unusual Nuremberg distinction: as 'free arts', considered part of the *artes mechanicae* (Summers 1987), they could avoid strict regulations and regulation as a closed trade. Within this system, Nuremberg sculptors sorted themselves out; they specialized in particular materials, which generally avoided direct competition and potential conflict (Kahsnitz 1986). Veit Stoss (1438/47–1533), for example, devoted himself to producing large, sculpted altarpieces, often polychromed, in limewood. Bronze casters in the city were led by the family workshop of

Peter Vischer the Elder (c. 1460–1529). Within this Nuremberg ecosystem of sculpture materials, then, Adam Kraft (ca. 1455/60–1508) chose to specialize in the less common material of stone, specifically gray sandstone, a material most closely associated with masons and builders, especially for durable major buildings, such as churches and city halls.

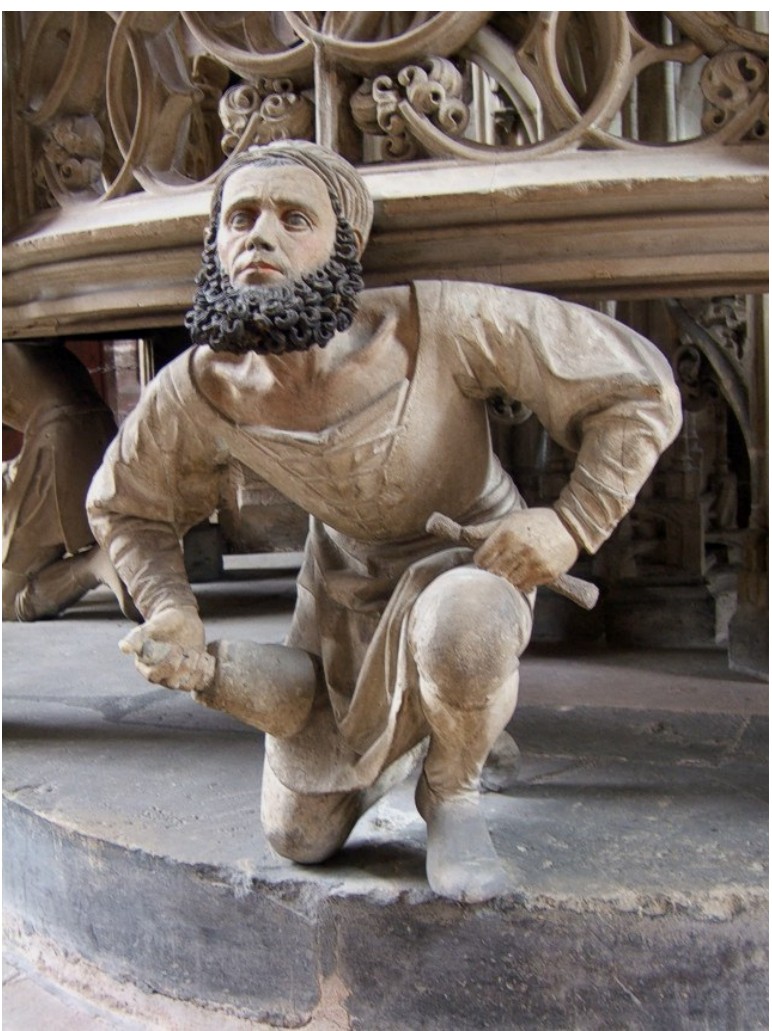

**Figure 1.** Adam Kraft, *Self-Portrait*, Sacrament Tower, St. Lorenz, Nuremberg, 1493–1496.

Stoss crossed over the agreed boundaries when he carved his own gray sandstone relief: the 1499 *Volckamer Epitaph*, placed prominently within the interior choir wall of St Sebald's. It was carved from a single block of the same material as the church building itself.[1] Reading conventionally from left to right, it presents the opening scenes of the Passion narrative: from the Last Supper through the Agony in the Garden to the Arrest of Jesus. These reliefs form a pious donation from a prominent local Nuremberg family. In similar fashion, Adam Kraft also received major patrician commissions in Nuremberg, especially for his monumental Sacrament Tower (1493–1496), commissioned by Hans IV Imhoff in the other major parish church of the city, St. Lorenz.[2]

Kraft's two major stone Passion monuments were also carved on patrician commissions. His *Schreyer-Landauer Epitaph* (1490–1492), placed on the apse exterior of St. Sebald's, was a joint commission from Nuremberg merchant Sebald Schreyer together with his nephew Matthias Landauer. Kraft's *Stations of the Cross*, his sequence of Passion episodes in six reliefs, was presumably made for a knight from Bamberg, Heinrich Marschalk von Rauheneck.[3] While the date of the latter work is undocumented, it probably postdates the well-documented St. Sebald's reliefs.[4]

## 1. Schreyer–Landau Reliefs

Located on the same site as a former external painted mural, already ruined by the elements, this carving was commissioned by Schreyer and Landau as carved replacements "to translate the subject matter of the painting into weatherproof stone" (1490–1492; Figure 2).[5] These reliefs, like Stoss's interior reliefs, function as an epitaph—a votive memorial with donors in perpetual prayer and hope of salvation as witnesses to a Gospel event, usually placed above a tomb or in a family chapel.[6] By convention, Kraft renders the family of donor figures in miniature, placed in kneeling prayer along with their families' heraldic coats of arms directly beneath the large-scale unfolding of events above them (Figure 3).

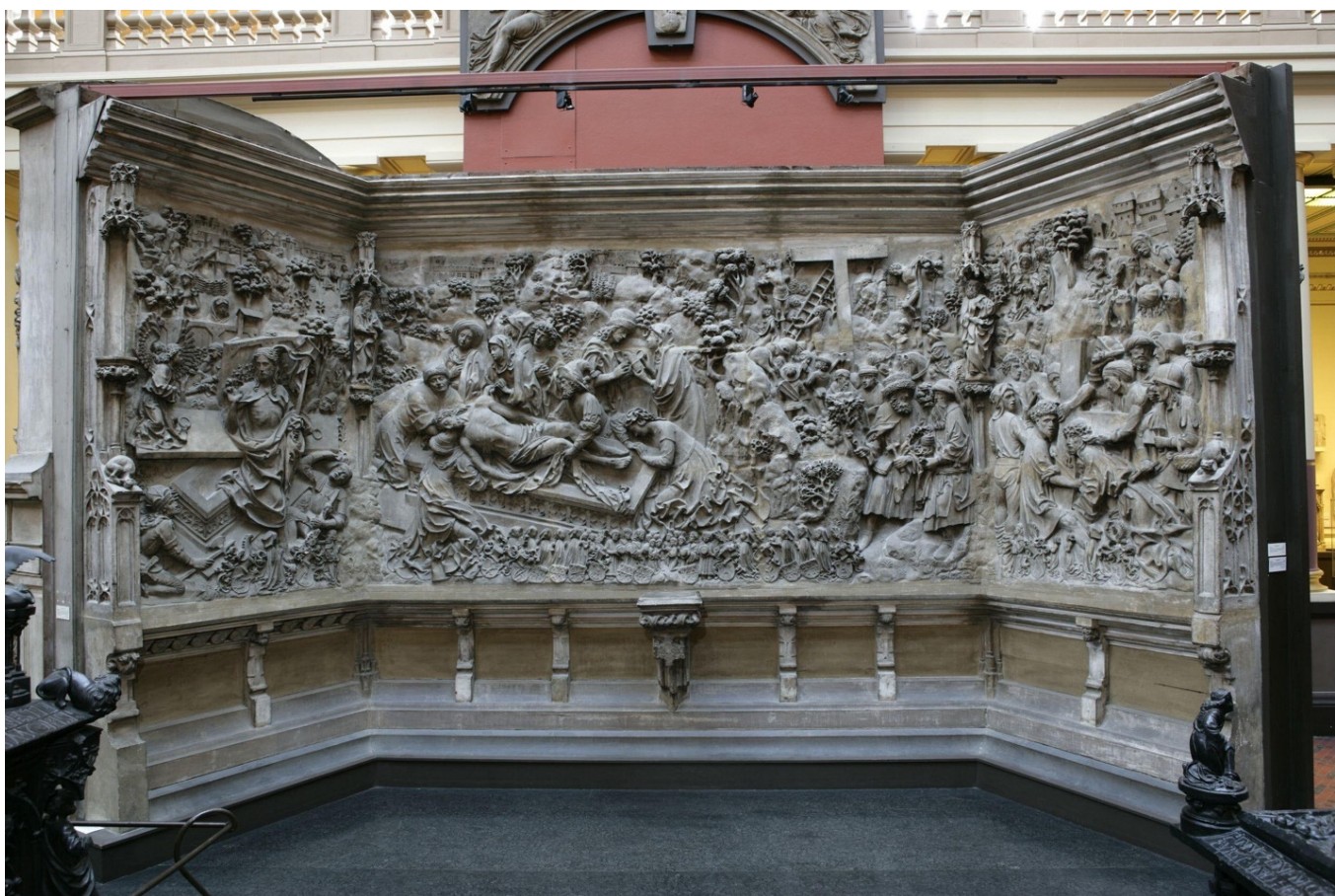

**Figure 2.** Kraft, Schreyer–Landau Epitaph, St. Sebald's, Nuremberg, 1490–1492 (Cast; London, Victoria and Albert Museum).

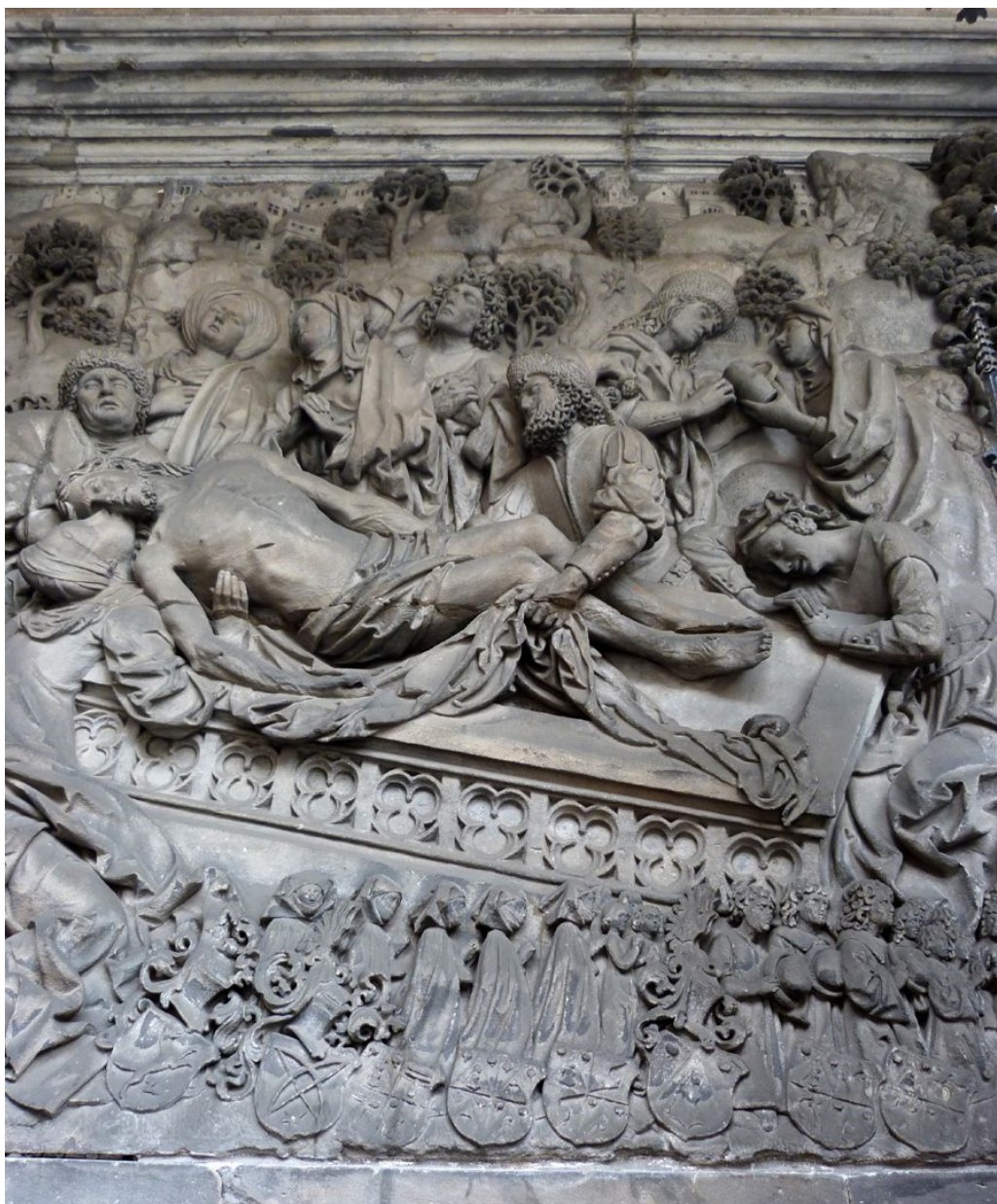

**Figure 3.** Kraft, Schreyer–Landau monument, detail: Donors and Deposition of Christ.

This arrangement closely parallels Dürer's own pair of painted epitaphs of Passion scenes, both painted around 1500: the *Lamentation* (c. 1500/03; Munich, Alte Pinakothek; Figure 4), painted for the goldsmith Albrecht Glimm, and the *Lamentation* (Nuremberg, Germanisches Nationalmuseum), a memorial for the prominent Holzschuher-Gruber family, also placed originally inside St. Sebald's.[7]

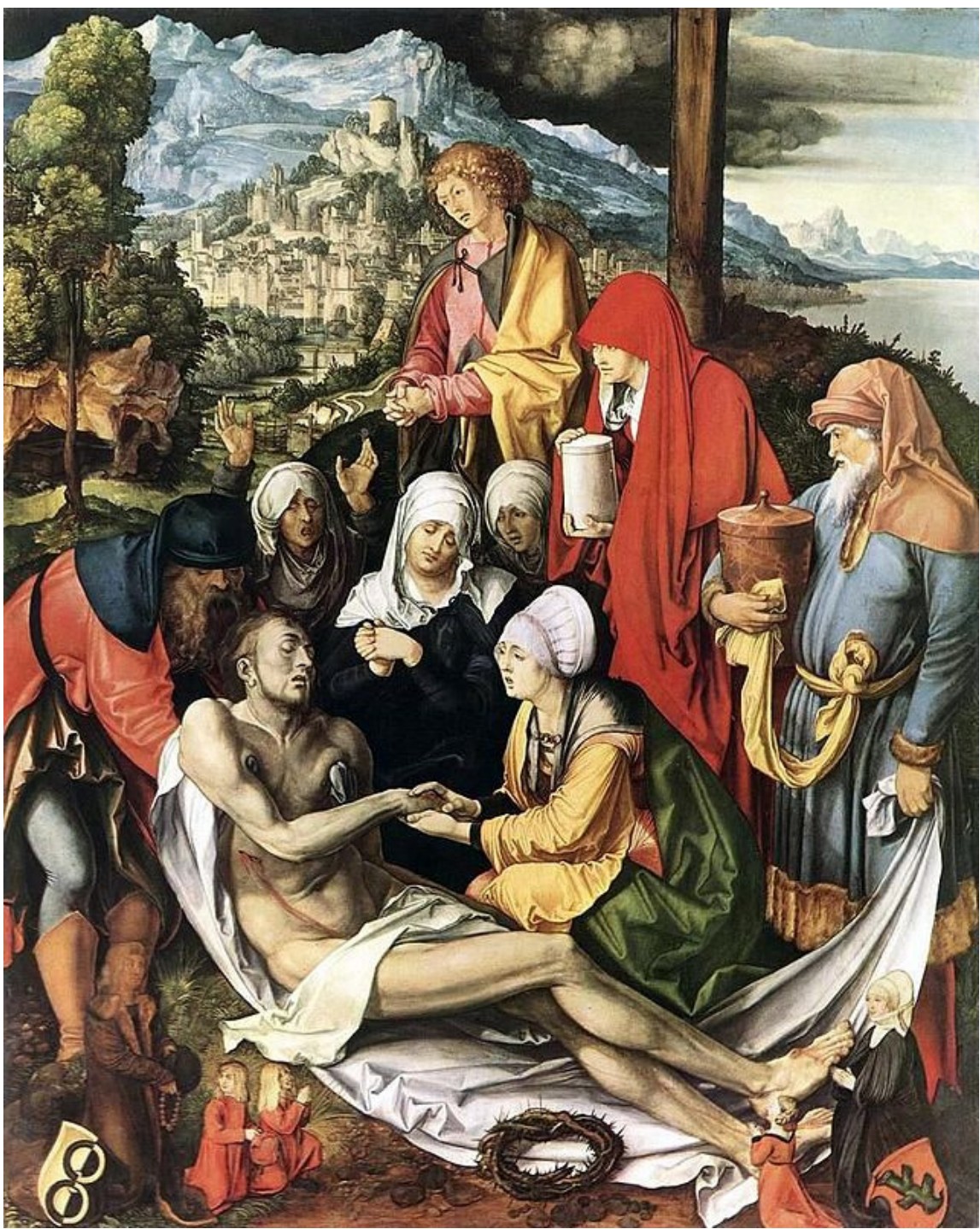

**Figure 4.** Albrecht Dürer, *Glimm Family Lamentation*, ca. 1500/03 (Munich, Alte Pinakothek).

In conjunction with the hope of salvation, so essential to the perpetual prayers of the donor family in an epitaph, Kraft's Shreyer–Landauer reliefs culminate in a Resurrection scene at the far left (Figure 5), even though chronologically this is the final Passion scene.[8]

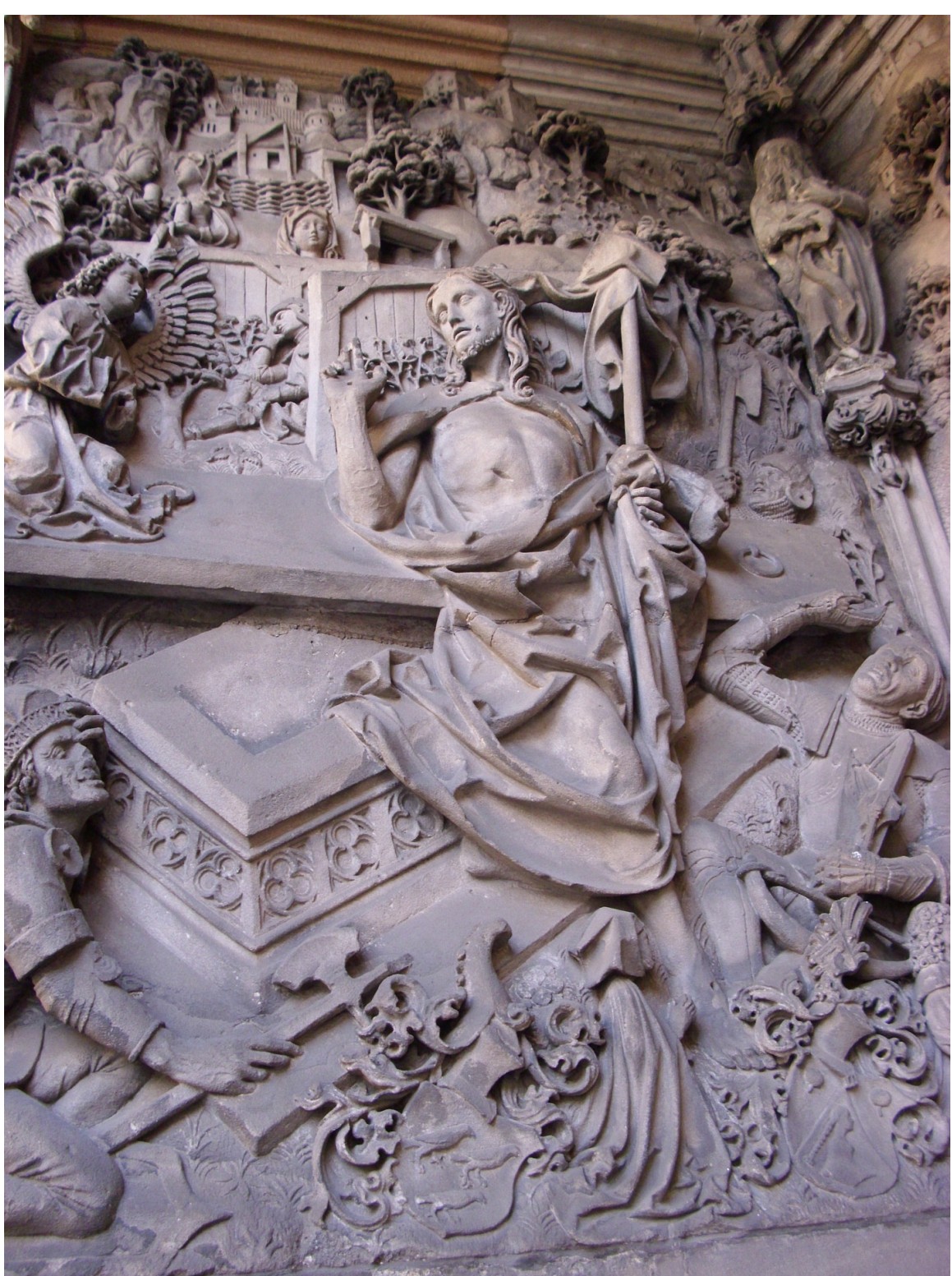

**Figure 5.** Kraft, Schreyer–Landau Epitaph, Detail: Resurrection.

Earlier Passion scenes begin at the far right with Christ Carrying the Cross (Figure 6), fallen to his knees while surrounded by mockers. The sympathetic holy figures, led by the Virgin, remain confined to a distant rise in the upper left.

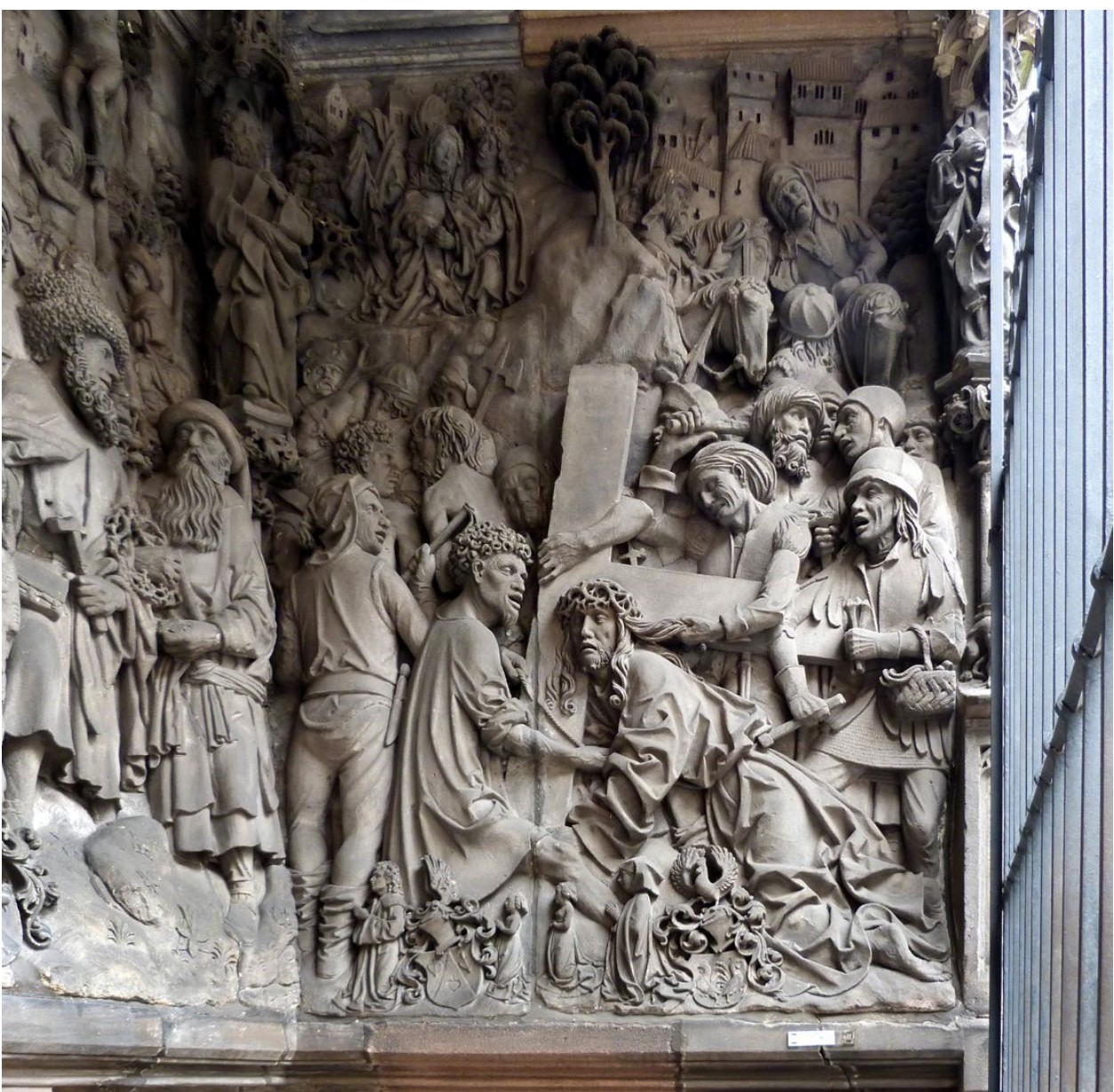

**Figure 6.** Kraft, Schreyer–Landau Epitaph. Detail: Christ Carrying the Cross.

Broad central Passion scenes follow, with the expansive Entombment of Jesus beneath the three crosses (that of Jesus is already empty). Here, a featured pair of relatively high-relief figures in late-fifteenth-century dress stride close to the viewer space (Figure 7).

One of them is clearly a self-portrait of Kraft himself as Nicodemus, holding tools appropriate to the Passion but also tied to his own craft of sculpture: hammer and pliers. The other man, holding the crown of thorns (identifying him as Joseph of Arimathea), is possibly Matthias Landauer.[9] Among the mourning disciples, two similar figures reappear, respectively, supporting the head and legs of the body of Jesus in the adjacent scene, the tomb setting of the *Deposition*. That moment extends the overall narrative continuity while remaining separate from the space where the two thieves still hang on their crosses. Further underscoring Kraft's pictorial elements, possibly deriving from the original lost murals, numerous background spatial details, such as city walls, trees, and hills, appear across all four scenes. Relief carving allows the sculptor to simulate the painted effects of the last murals while still working in his specialty, the more durable material of sandstone.

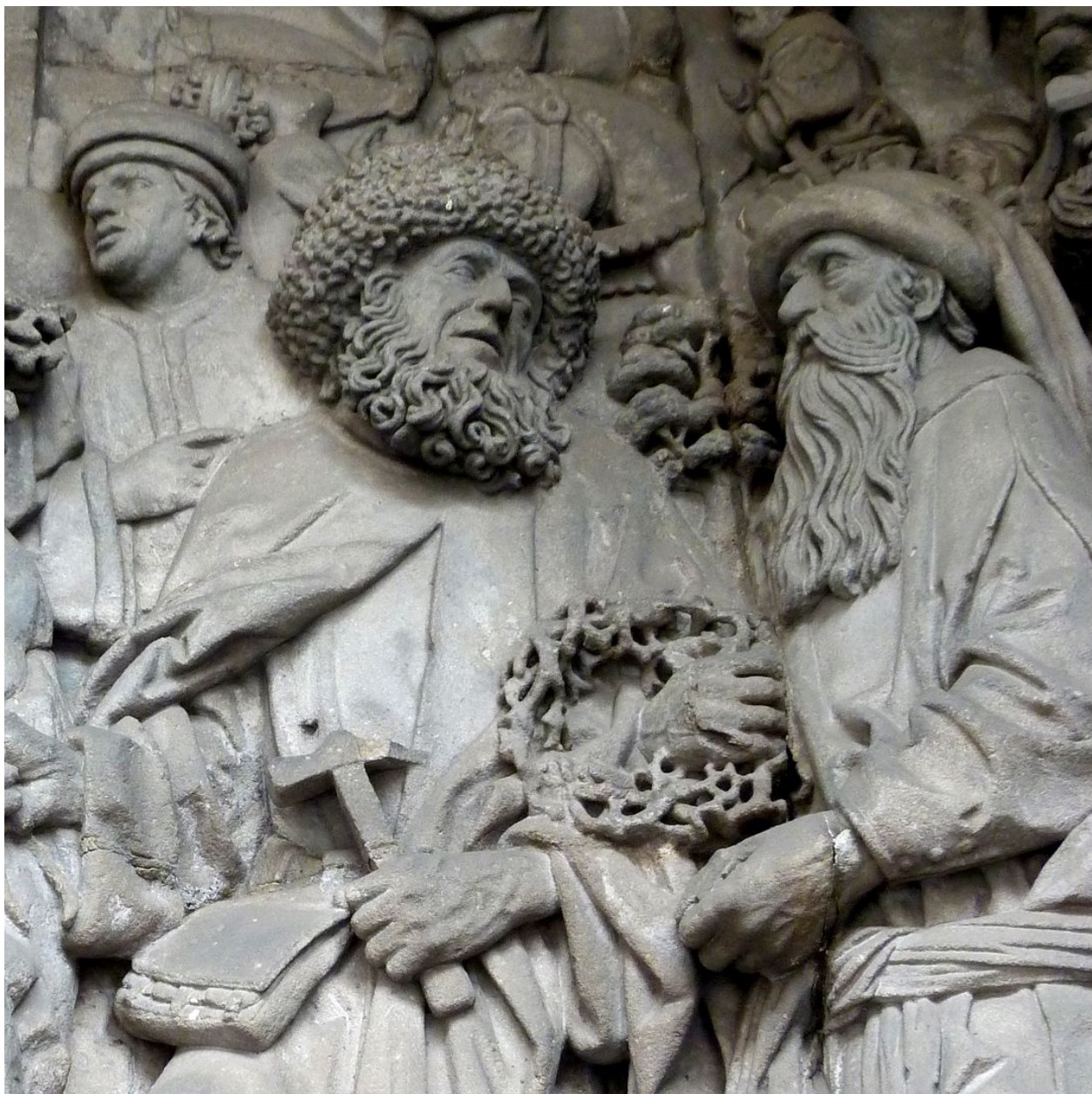

**Figure 7.** Kraft, Schreyer–Landau Epitaph, Detail: Nicodemus (Self-Portrait) and Joseph of Arimathea from *Deposition*.

Of course, in his prints, both woodcuts and engravings, and in two sets of drawings (the *Green Passion*, 1503/04, Vienna, Albertina; *Oblong Passion*, 1521–1524),[10] Albrecht Dürer favored an unfolding presentation of the Passion as a sequential visual experience, and in this graphic unfolding, he was already following the late-fifteenth-century precedent set by Martin Schongauer.[11] In every case, the movement of individual images runs from left to right (Figure 8), and as the sequence of folios proceeds, like the turning pages of a book (or a modern graphic novel), so do they also move in time from the leftmost to the rightmost.

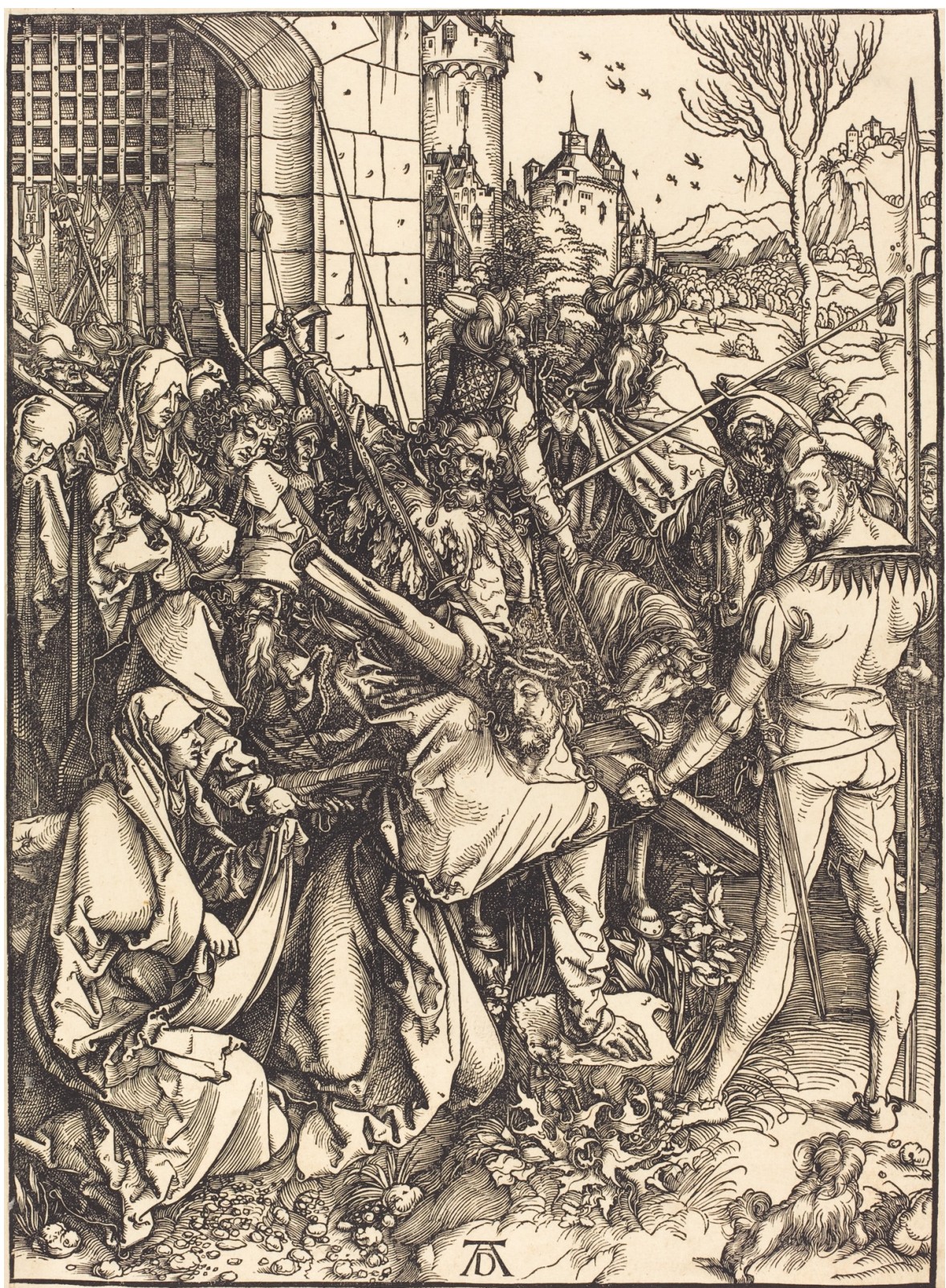

**Figure 8.** Albrecht Dürer, Christ Carrying the Cross from Large Woodcut Passion, ca. 1498–1499.

In the case of his two bound woodcut cycles, that narrative progress was expressly reinforced by the process of turning the pages of a bound volume, and Dürer follows viewer habits of eye movement from left to right in his isolated version of the Carrying of the Cross (Figure 9). Of course, the earliest Dürer Passion cycles were created after Kraft

had already fashioned his own epitaph reliefs on the outside of Nuremberg's busiest parish church, which was quite close to Dürer's house.[12] Regardless of the direction that one took to pass along the apse of St. Sebald's, to follow Kraft's carved narrative sequence of the Passion, one had to begin at the far right with the Carrying of the Cross and proceed across the Entombment to end at the left with the Resurrection.

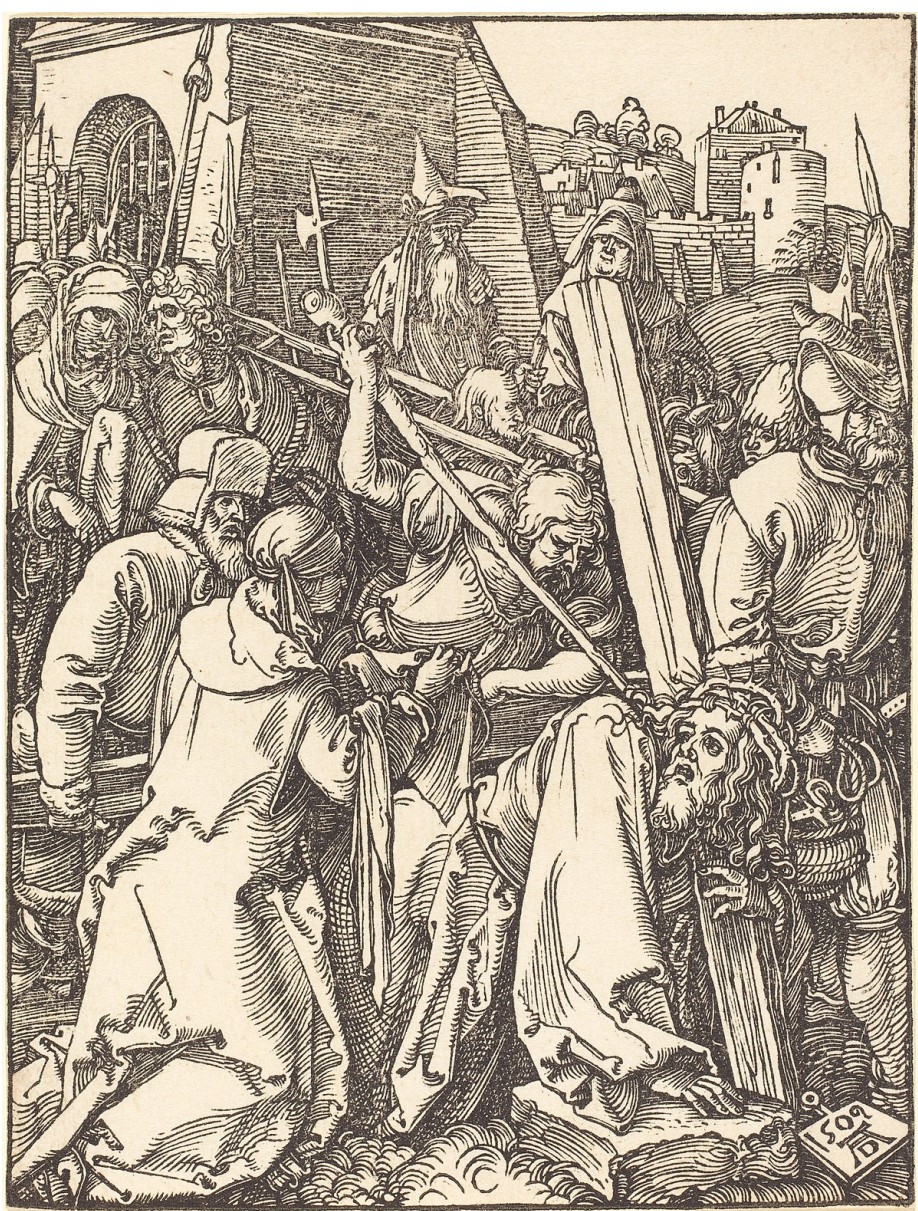

**Figure 9.** Albrecht Dürer, Christ Carrying the Cross from Small Woodcut Passion, 1511.

Late-medieval piety often sought to appeal to viewer emotions, especially in Gospel narratives—whether amplified in prose retellings with vivid added details or in details of visual presentations.[13] Certainly within this affective emotionality, the Passion story loomed large, and the violent torments of Jesus were elaborated and multiplied to emphasize his physical suffering. Also included in that Passion imagery—and consonant with the accompanying texts with Dürer images, especially the Latin verses by Benedictus Chelidonius within the 1511 *Small Woodcut Passion*—was a consistent, virulent anti-Semitism, conveyed by Dürer through the grotesque faces, mocking gestures, and outright torturing inflicted by Jewish opponents of Jesus.[14] That recurrent anti-Semitism would resurface even more emphatically in Kraft's later series of reliefs for the Way of the Cross.

These Jewish tormentors are further distinguished by their dress markers, especially the hats of their leaders, whether alien turbans or the pointed *Judenhut.* The poem by Chelidonius on the title page of the 1511 *Large Woodcut Passion* concludes: 'Let it be enough that I endured such great torture/Under the Jewish foe. Now, my friend, grant me peace.'[15] In the title pages of both Dürer woodcut *Passion* volumes, issued in the same year, 1511, Jesus as the Man of Sorrows is seated on a cold stone; the *Large Woodcut Passion* confronts him with a hostile tormentor who proffers a reed mock scepter.

Kraft's St Sebald reliefs, particularly his *Christ Carrying the Cross*, show a similar confrontation, plus a dense cluster of other grimacing faces around Jesus with the cross. Even the sympathetic figures of Nicodemus and Joseph of Arimathea nearby wear distinctive Jewish costumes, especially headwear.

## 2. Stations of the Cross Reliefs

But Kraft was not finished with the Passion narrative after St. Sebald's. Along the pathway from the walls of Nuremberg to the St. John Cemetery, the sculptor fashioned seven, over-life-sized, sandstone slabs on pillars with large sandstone reliefs (122 × 165 cm). These images, culminating in an image of the Deposition that shows only the holy figures, track the footsteps of Jesus on the Stations of the Cross, the Via Dolorosa in Jerusalem. The 1508 date of completion, shortly before Kraft's death, was already reported before the mid-sixteenth century (1546) by local historian Johann Neudörffer (although, as noted, that dating has also been contested and reasserted instead as an early work, around 1490). Despite the absence of any coat of arms or dedication, the patronage is often assigned to Ritter Heinrich Marschalk von Raueneck (d. after 1519), who also commissioned a similar Stations of the Cross in his nearby hometown of Bamberg. Another, late-seventeenth-century account assigns patronage to a different candidate, the Nuremberg Jerusalem pilgrim Martin Ketzel.[16] Neither claim has solid documentation, so the details of the commission remain uncertain.

Kraft's sequence originally culminated at the St. John's Cemetery in a large-scale carved *Calvary* ensemble with the three crosses, preserved only in reworked fragments, but also recorded in later print images (Kraft's *Christ on the Cross* is preserved in Nuremberg's Heilig-Geist-Spital; Figure 10).

Also located at the cemetery, the Holzschuher Chapel (dated 1508; Figure 11 offers a sculpted *Entombment* in high relief, virtually in the round, by Kraft. It stands in front of a painted bird's-eye image of Jerusalem that heightens the accuracy of the imagery and suggests the experience of an actual pilgrim. Of course, this three-dimensional finale could still have been commissioned well after his *Stations of the Cross* reliefs, so the date of the overall ensemble, like the patron, remains contested.

Even after the definitive loss in 1187 of Jerusalem and in 1291 of the Latin Kingdom to Islamic forces during the Crusades, the Holy Land had remained a desired goal of pilgrimage and meditation, simulated in Europe in a variety of structures with claims to replicate the Holy Sepulchre in particular.[17] Within medieval Europe, Franciscans developed a sequence of the Fourteen Stations of the Cross as a meditational practice. In turn, that pious contemplation relates to other vivid contemporary prose narrative retellings of Gospel events for individual readers, in such 14th-century texts as Pseudo-Bonaventure's *Meditations on the Life of Christ* or Ludolph of Saxony's *Vita Christi*, as well as in Passion plays.[18] Its meditational sequence also overlaps with late-medieval cults around the Passion, such as the Seven Falls of Jesus or his Five Wounds or related clusters of devotion: the Seven Sorrows of the Virgin and the *arma Christi*, or instruments of the Passion.[19] The Stations of the Cross truly were more connected to episodes in the Gospels than to actual sites in Jerusalem; only later in the early modern period were specific locations in Jerusalem codified as the stops on the Via Dolorosa.

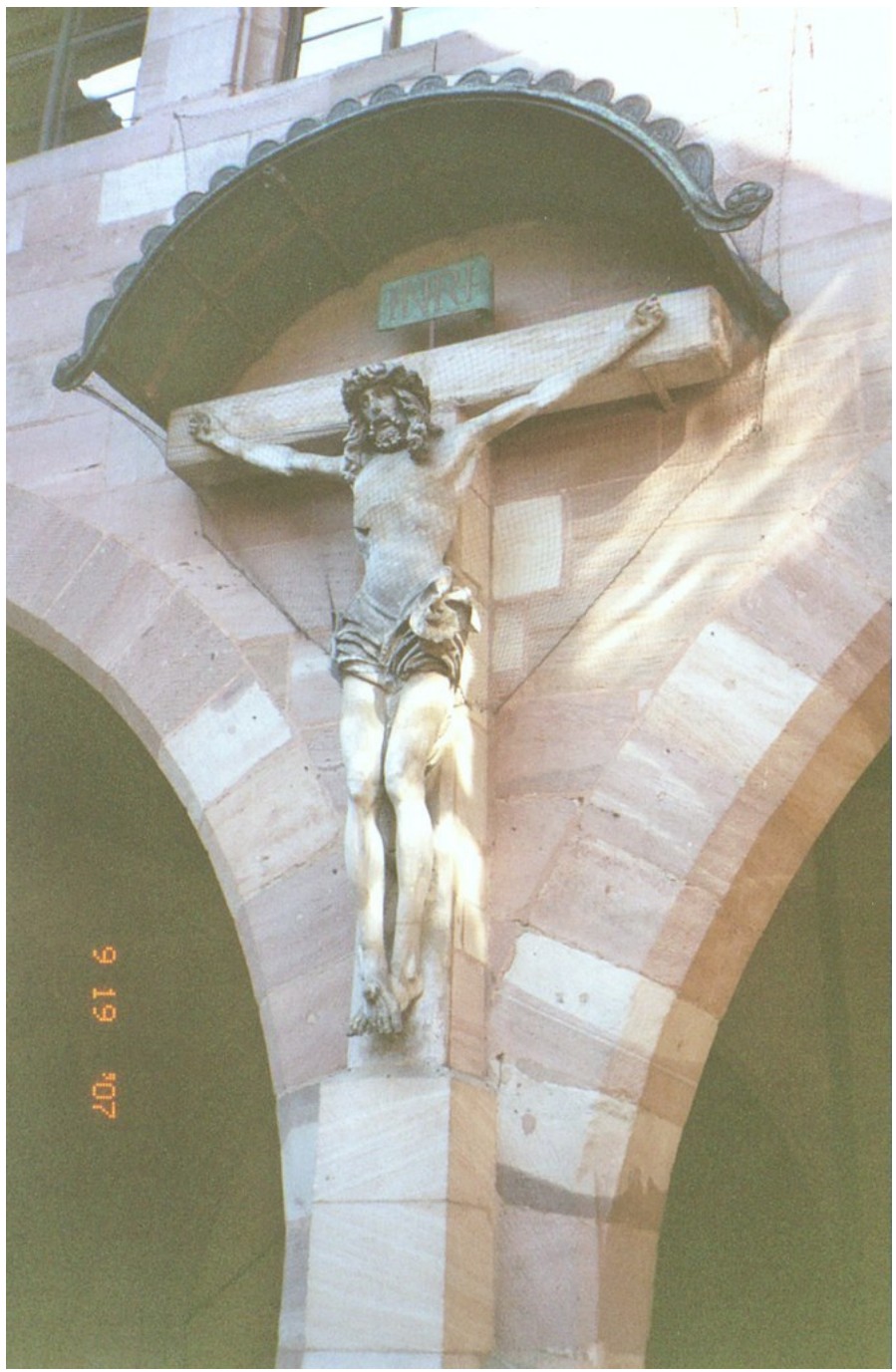

**Figure 10.** Kraft, *Christ on the Cross*, from *Stations of the Cross*, ca. 1506–1508. Fragment (Nuremberg, Heilig-Geist Spital).

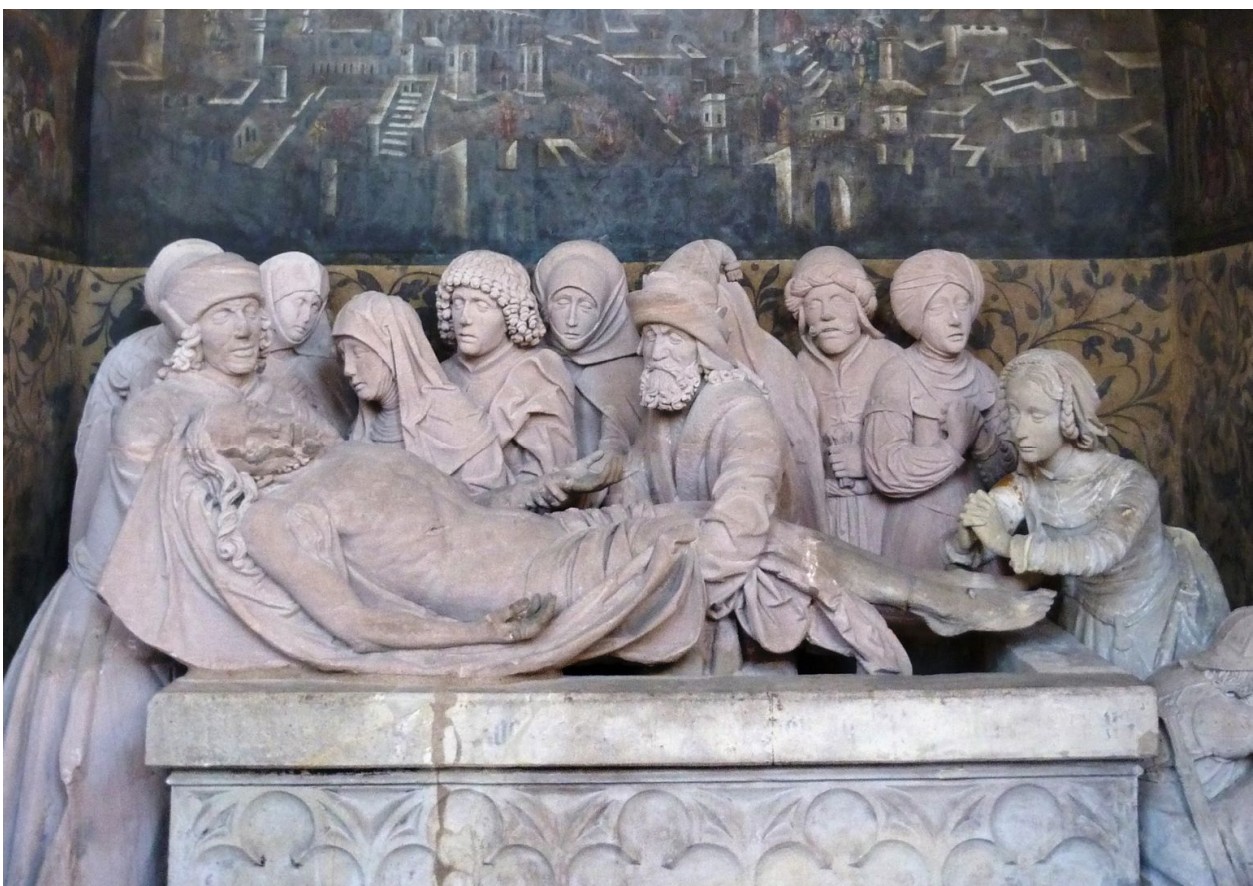

**Figure 11.** Kraft, *Entombment* from *Stations of the Cross* (Nuremberg, St. John's Cemetery, Holzschuher Chapel), 1508.

However, after Franciscans gained control of access to Jerusalem sites for actual pilgrims in 1312 under the reign of Pope Clement V, they were able to encourage pilgrims to visit specific individual sites—or their reproduced simulacra in Europe, particularly of the Holy Sepulcher—closely associated with papal indulgences.[20] Viewing the actual sacred places traversed by Jesus on his way from the house of Pilate to Golgotha soon became fixed practice, which was associated with the actual spiritual benefit of contemplating the Stations of the Cross. Within the focused late-medieval devotion focusing on Christ's Passion, such commemoration and identification with both the process and procession to Calvary remained paramount in the experience of the devout. For example, pilgrimages to Jerusalem (1458, 1462) by William Wey already use the terms *stationes* and *Via crucis/Via dolorosa* for the principal stops along the route of Jesus.[21]

Simultaneous visual incorporation of events within a simulated bird's-eye view of Jerusalem was already represented by Netherlandish painter Hans Memling in his panoramic *Passion* (ca. 1470/71; Turin, Galleria Sabauda; Figure 12). It features a foreground emphasis on the Carrying of the Cross, where Christ's outward gaze summons the most empathic appeal to a meditating viewer.[22]

Well after Kraft's lifetime, the 1584 Jerusalem map (Figure 13) by Dutch author Christian van Adrichom (1533–1585), part of his atlas and description of the Holy Land, fixed the number of sites at fourteen for the Stations of the Cross and keyed those places onto the more exact mapped topography of the city and its major sites, which were numbered and keyed to a descriptive text.[23]

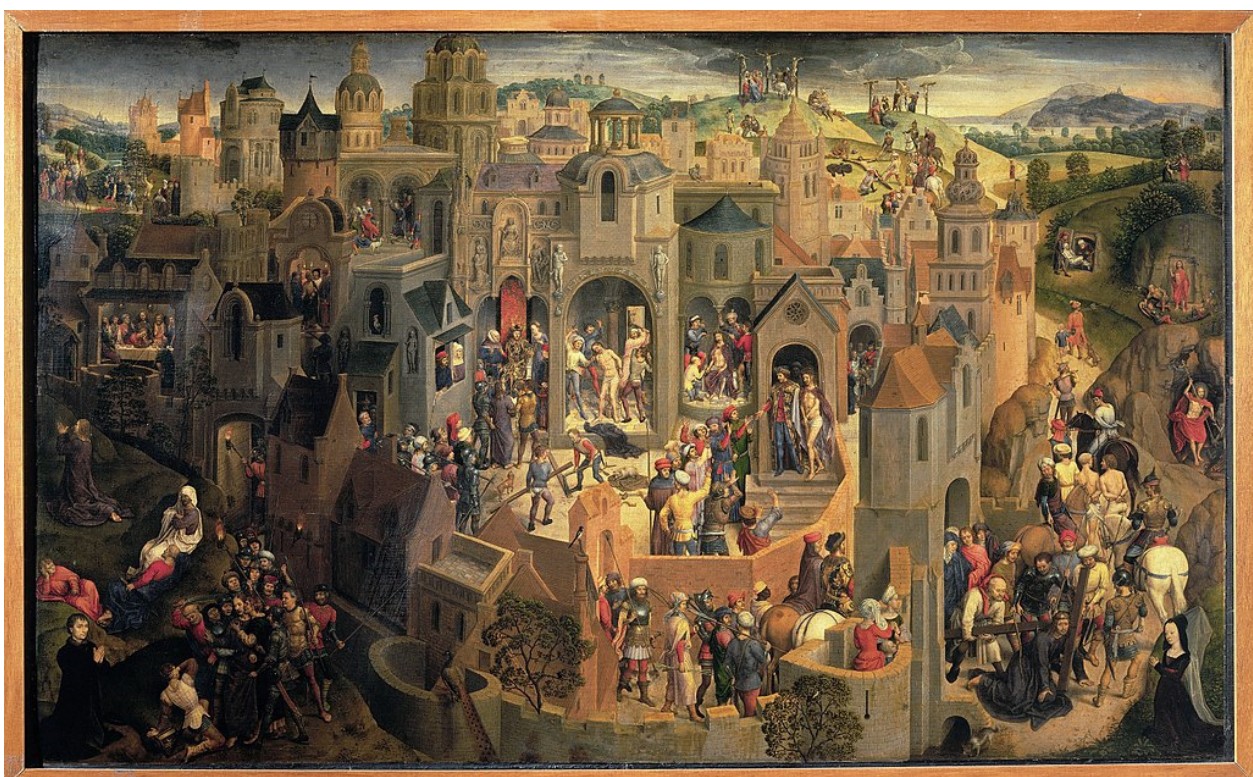

**Figure 12.** Hans Memling, *Panorama with Passion of Jesus*, ca. 1470/71 (Turin, Galleria Sabauda).

Thus, when Adam Kraft turned his attention to his seven carved, large-scale reliefs of the Way to Calvary, he had ample precedents to draw upon and numerous models for individual scenes, both in previous print cycle illustrations and even earlier stone reliefs.[24] Kraft's reliefs contain vernacular German inscriptions that not only describe the depicted Passion moments but also number the precise steps between each event down the *Via crucis*, starting from the home of Pilate. Such exactitude and authenticity to the original events and topography assumed primary importance for the devout. The same precision also measured other Passion details in both contemporary texts and images: the tomb's length; the body size of Jesus's; his physical likeness (as described in a spurious 'Lentulus letter' or imaged in the relic of Veronica's cloth); and even the actual size of his side wound, which was depicted at full scale in both prayerbooks and prints.[25]

By starting at Nuremberg's own city walls and proceeding westward to its external St. John's Cemetery (Figure 14), the pathway ornamented with scenes by Kraft transforms the German city of the sixteenth-century present into biblical Jerusalem for all devout pedestrians who follow his simulacrum of the Via Dolorosa. Certainly, numerous prominent Nuremburg patricians, including several members of the Tucher family, made their own pilgrimages to Jerusalem itself and joined pious brotherhoods devoted to the Holy Sepulchre or similar devotions.[26] A chart of the route by Hans Tucher survives with accounts recording the exact number of steps between stops. In addition, the Tuchers commissioned a painted epitaph for St. Sebald's from Master LCz with the subject of *Christ Carrying the Cross* (1485), surrounded by figures during his procession toward Calvary, extending out from the city walls at right.[27]

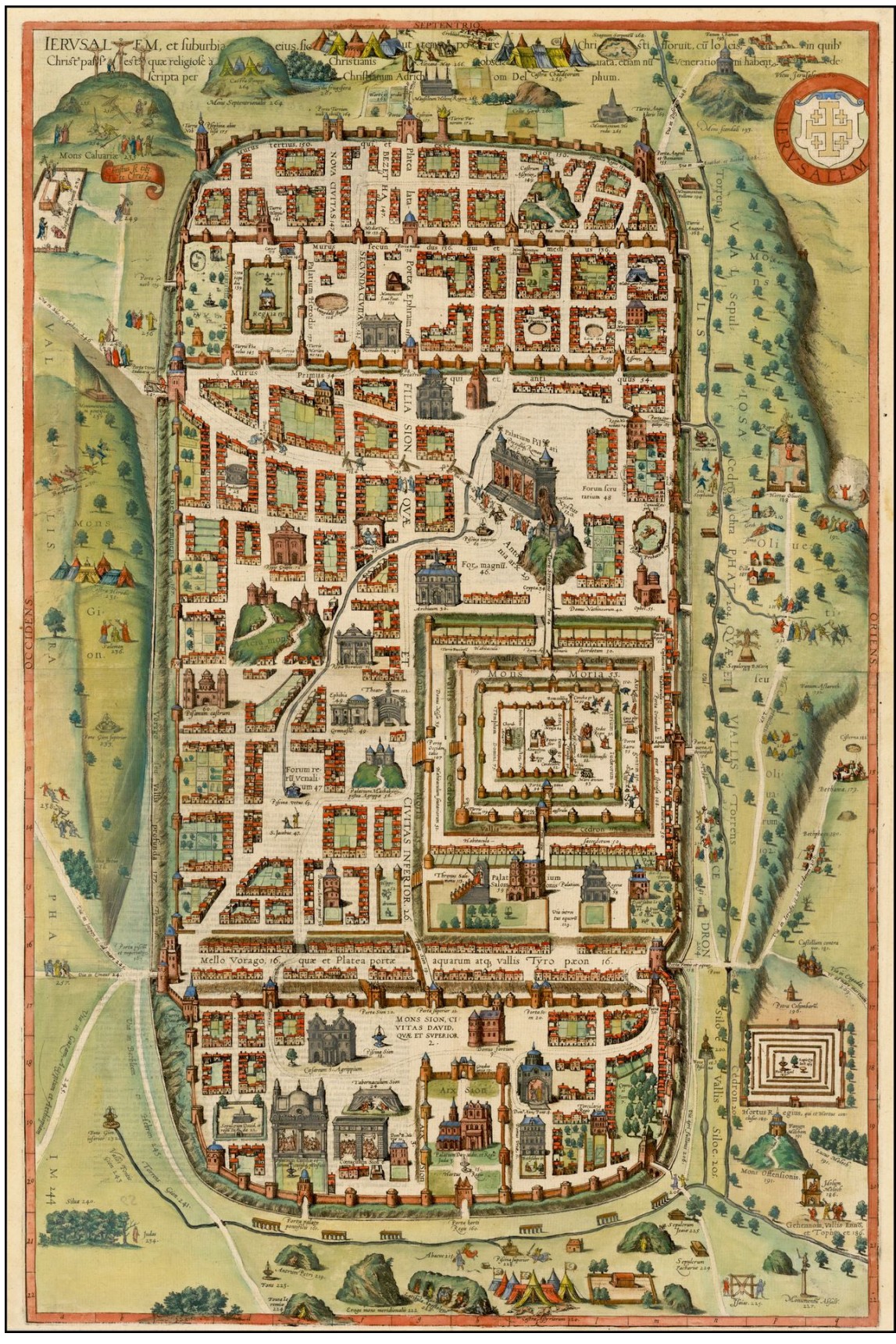

**Figure 13.** Christian van Adrichom, *Map of Jerusalem with Via crucis*, from Georg Braun and Frans Hogenberg, *Civitates orbis terrarum* (Cologne, 1590).

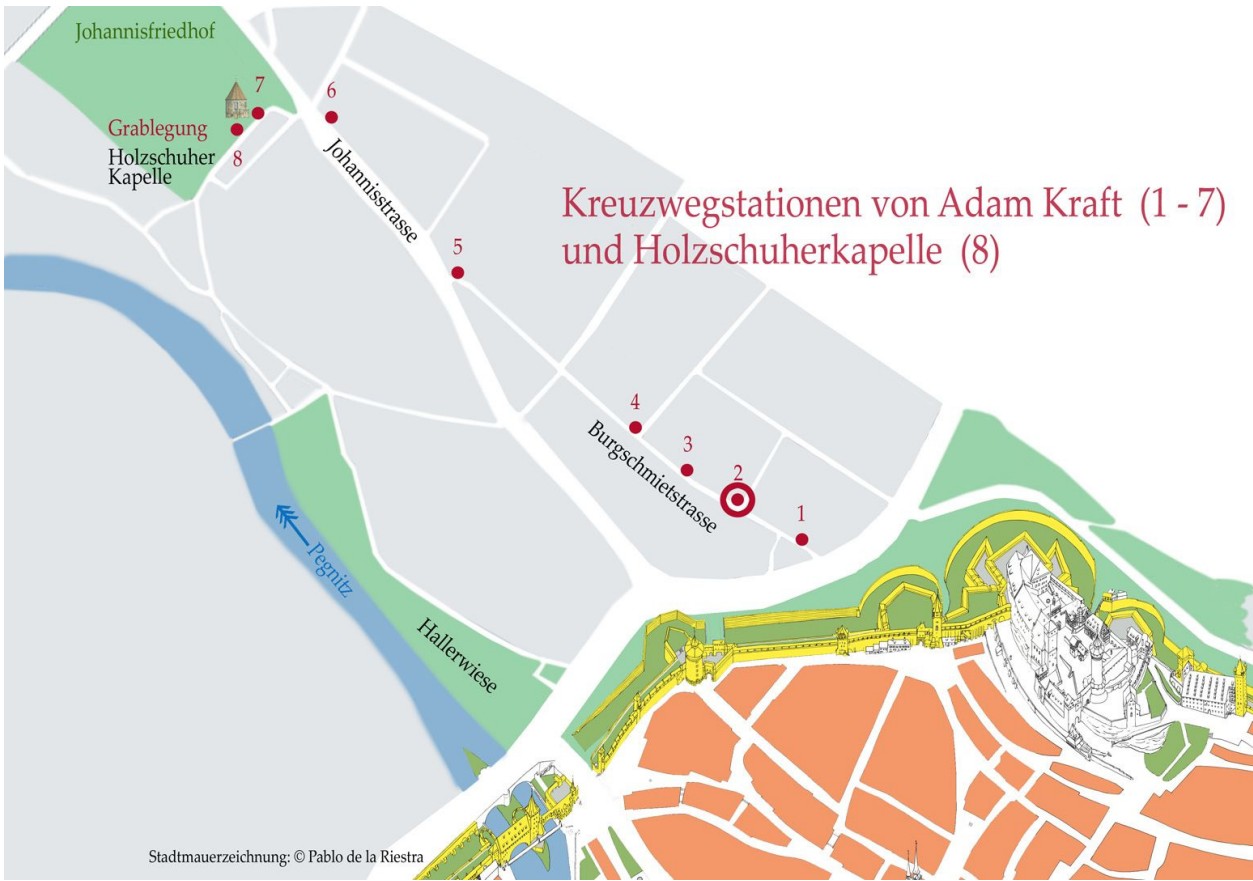

**Figure 14.** Map of Kraft, *Stations of the Cross*, 1506–1508 in Nuremberg.

All Kraft reliefs appear in standard size at a slightly horizontal format. Figures overlap slightly, standing at full length, their heads near the top of the frame. Kraft renders their bodies and movement sensitively beneath the folds and surfaces of their garments. With no backgrounds, figural overlap suggests depth in space. In terms of evocative emotions, Kraft vividly contrasts the grotesque faces and vicious gestures of the tormentors against Jesus's own patient resilience, often witnessed sympathetically by the groups of somber compassionate disciples and holy women.

In the first relief (Figure 15), Jesus appears, already bearing the heavy burden of his cross, which reaches up to the top of the frame in the center. His movement proceeds leftward, once more positioned against the habitual grain of reading from left to right, which suggests his slow, heavy pace under the burden of the cross along with the resistance imposed by his abusive enemies. In this one relief, his agony is enhanced by the horizontal drag of a spikeblock attached to his foot, although its nails are no longer visible.[28] The figure of Jesus is the most fully rounded individual, so he sets the pictorial foreground. Behind him, a tormentor, seen from behind, also implies spatial depth from the other layers of figures whom he overlaps (much like the figures of Giotto in the Arena Chapel). In the first scene, Jesus is juxtaposed with a cluster of his followers at left, led by the Virgin Mary in the foreground, plus St John the Evangelist and other supporting figures. They provide a visual pause in the mournful progress of the cross. The bent figure of Jesus as prisoner is surrounded by a row of abusive figures, including the one behind him, who lifts his arm to strike. While on the original block (Nuremberg, Germanisches Nationalmuseum) the faces of these figures are now weathered, and that of Jesus fully lost except for his flowing locks, several of these tormentors show open mouths, seemingly for jeering shouts. Their faces, boots, and headgear vary for each individual. In contrast with the disciple group, these hostile figures display large noses and hostile stares like the tormentors in Passion prints by both Schongauer and

Dürer. The vernacular German text of the relief declares: 'Here Christ meets his worthy, beloved mother, who swoons from great heartache. 200 steps from the House of Pilate.'[29] Thus, the measurements of distance between the original Jerusalem sites of the Via Dolorosa are expressly invoked on each successive relief.

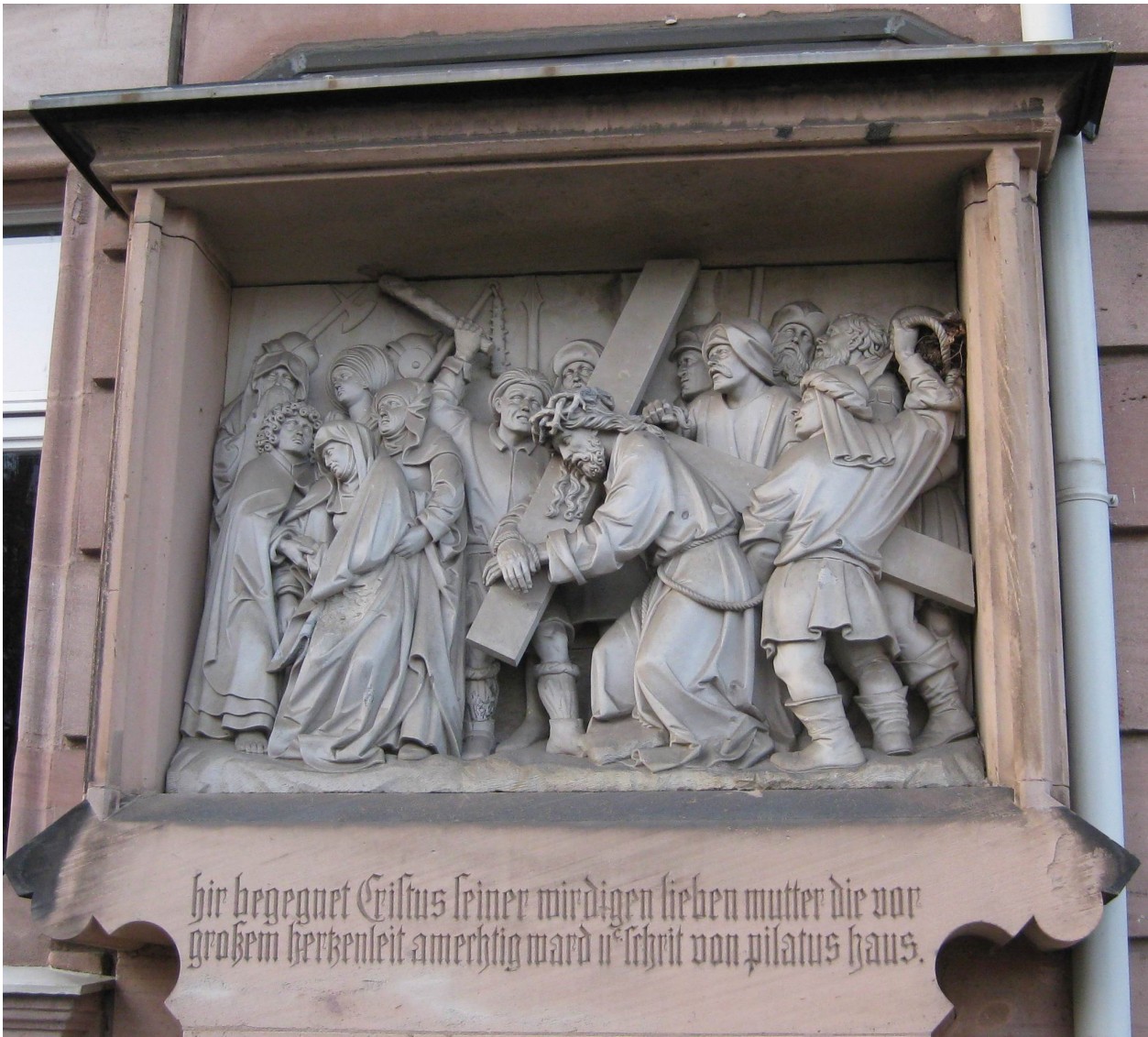

**Figure 15.** Kraft, *Stations of the Cross*, Nuremberg, 1506–1508. Station One (reconstruction).

Kraft's second relief (Figure 16) shows the familiar episode when Simon of Cyrene is drafted to share the burden of the cross (Matthew 27: 32; Mark 15: 22; Luke 23: 26). Several of the same ugly heads reappear here among the tormentors, who are now three deep. They completely surround the cross and Jesus, who is now positioned slightly left of center. The overall movement still remains right-to-left, as Christ's knees still buckle, despite his added assistance with his burden. Two men press Simon onto the diagonal cross, while a man in armor between Simon and Jesus tugs at his hair to add further torment and delay. At the front of the procession, a backward-facing man, seemingly tugging at Jesus with a rope, bends backward and almost breaks out of the left frame. The inscription reads: 'Here Simon is compelled to help Christ carry his cross. 295 steps from the House of Pilate.[30]

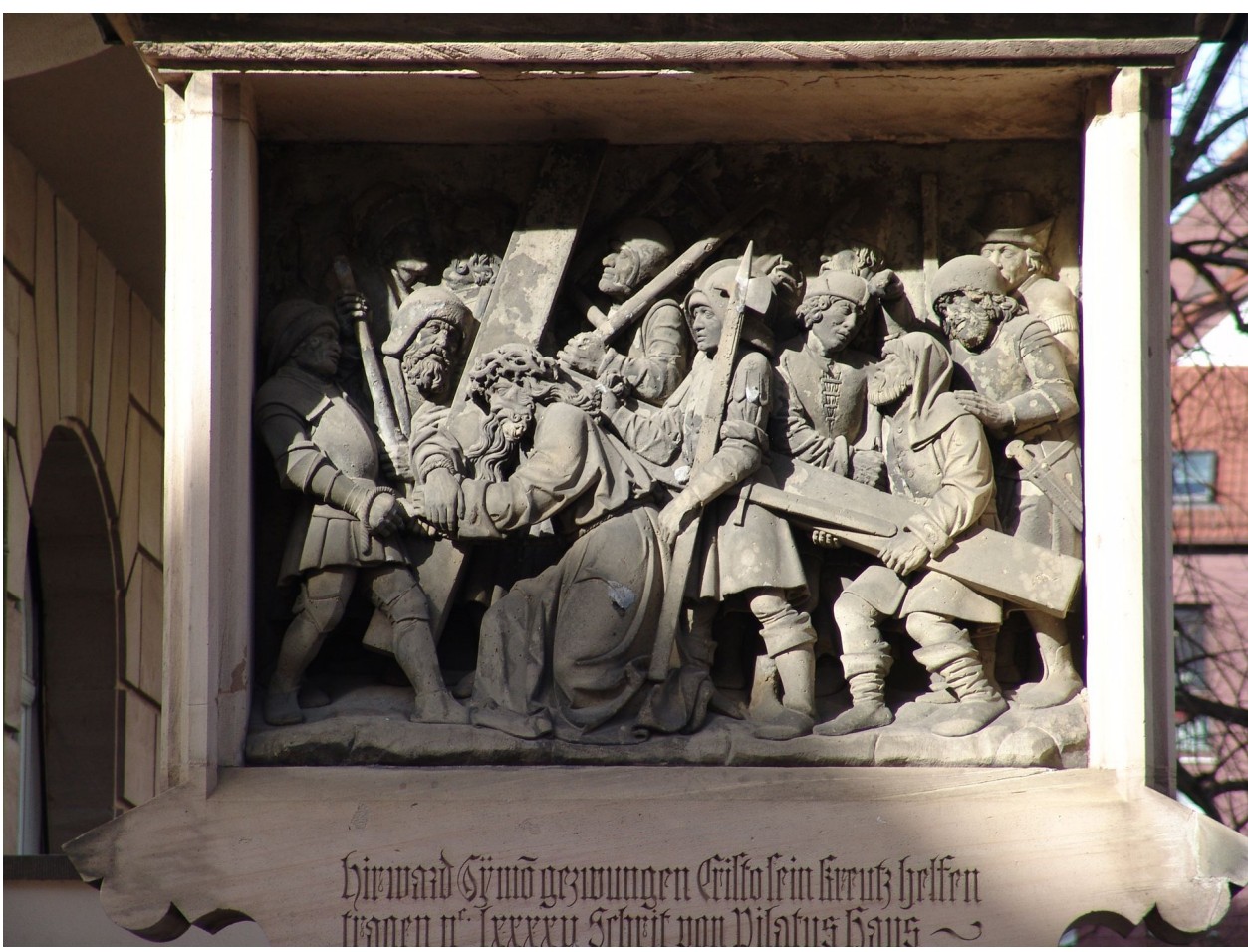

**Figure 16.** Kraft, *Stations of the Cross*, Nuremberg, Station Two (reconstruction).

The third relief (Figure 17) inverts the first one, with two groups turned toward a central Jesus with the cross. Again, Kraft juxtaposes open-mouthed tormentors, now at viewer left, against the quiet, mourning women of Jerusalem in three layers at right. Jesus, on his knees in the center, turns his head rightward in response to the women, uttering words of comfort. Both his flowing hair and the crown of thorns frame his fine features. This image starkly contrasts the hostile, mobile males (one of them, closest to the prisoner, tugs his hair) with the holy women. All the men wear varied headwear and short tunics. Behind this mad group, the modest cluster of women all face Jesus with clasped hands and covered heads; they are dressed in uniform, full-length robes. Even Simon of Cyrene is shouting at Jesus as he helps bear the cross. Yet one man in a turban continues forward in profile at the left edge, ignoring the halt in the procession behind him. Ultimately, these holy women model proper behavior for those faithful in the process of re-enacting the Passion procession, as they evoke compassion for the suffering Jesus instead of its cruel opposite.[31] The inscription defines the scene: 'Here Christ spoke, "You daughters of Jerusalem, do not weep over me, but over yourselves and your children." 380 steps from the House of Pilate.'[32]

Kraft's fourth station (Figure 18) is the most damaged, but it is also a crucial scene of personal encounter: St Veronica meets with Jesus, whose face miraculously imprints on her veil. Jesus still remains at the center of the image, in high relief, his face in profile before the cross, while Veronica faces him from beneath a gateway at left and presents the holy relic, the Vera Ikon, to him (Hamburger 1998). At the far right stands an armored soldier, seen from behind, and a second soldier with a sword follows the cross immediately behind Jesus, but he has lost both his head and legs. Facing Veronica, an additional remarkable round face with a cap emerges from behind the cross, giving physical depth as well as emotional counterpoint

to the moving encounter between the two principals, both saint and Christ. Its inscription identifies the event: 'Here before her house Christ has impressed/ printed [*gedruckt*] his holy face for holy woman Veronica on her veil. 500 steps from the House of Pilate.'[33]

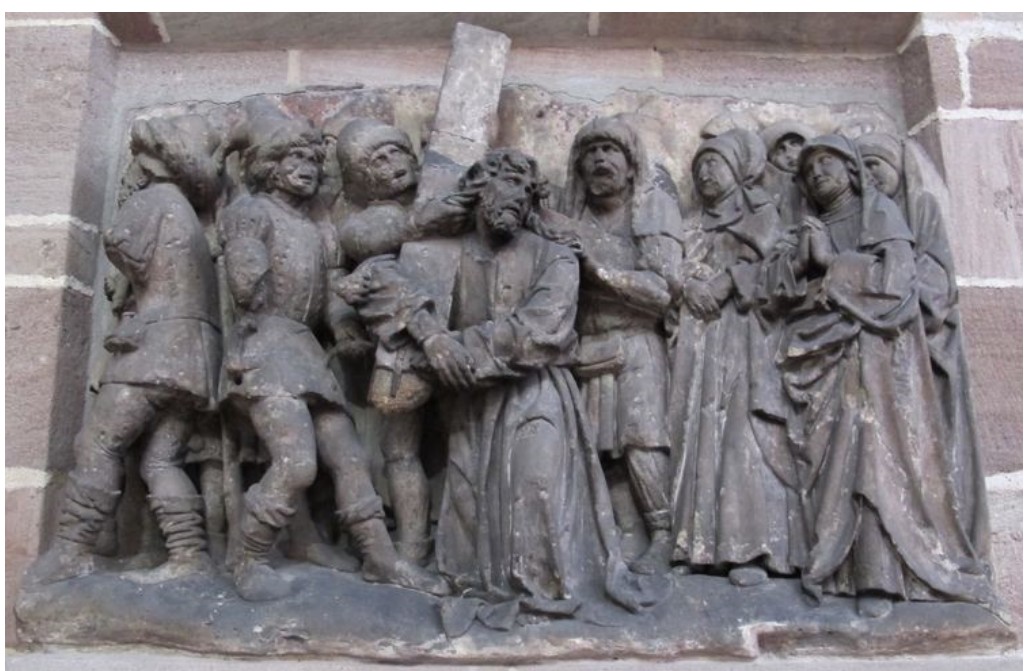

**Figure 17.** Kraft, *Stations of the Cross*, Nuremberg, Station Three (Nuremberg, Germanisches National­museum).

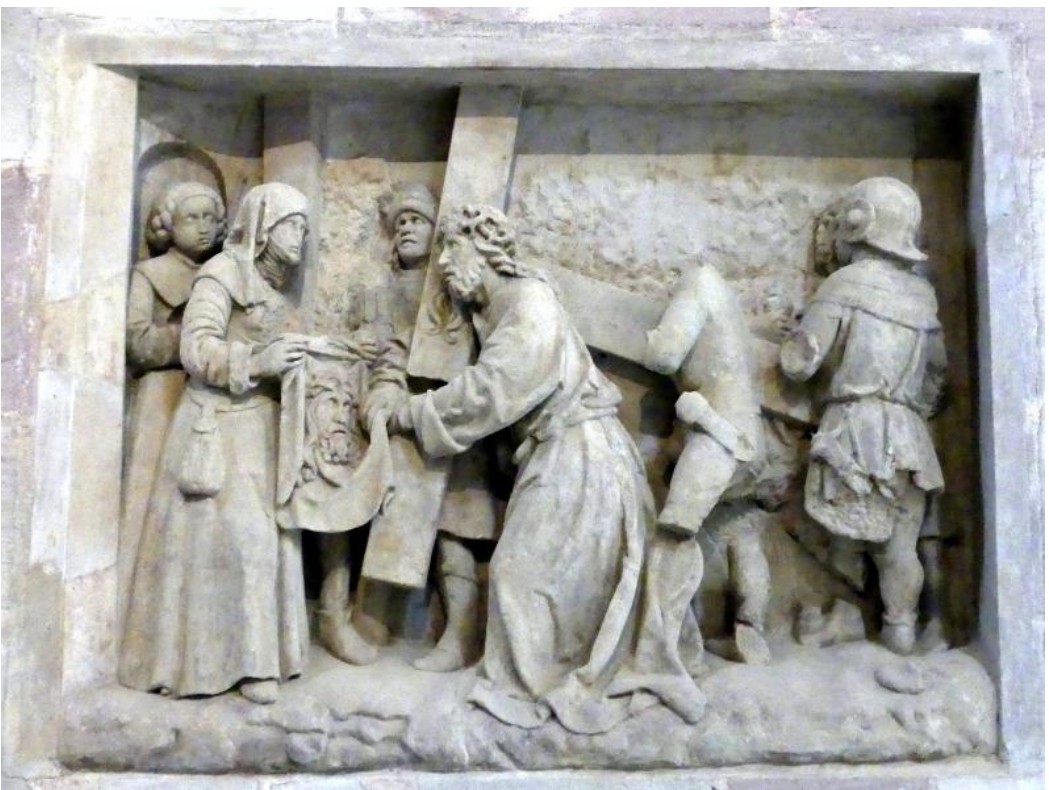

**Figure 18.** Kraft, *Stations of the Cross*, Nuremberg, Station Four (Nuremberg, Germanisches National­museum).

The fifth relief (Figure 19) is densely filled, consisting exclusively of Christ's enemies, who assail him from both sides. His hands overlap on the cross while bound with rope. This preserved format is currently square rather than the original horizontal breadth of other works in the sequence, and it has lost a figure at its far right, so it is represented here by the reconstruction. The actors appear on three levels, with the three foremost figures in high relief, such that the leftmost figure projects beyond the frame itself, almost into the actual space of the beholder. One unusual new figure appears at right: a forward-facing, long-bearded man with a distinctive high cap. While the inscription here is more general, it is expressly anti-Semitic: 'Here Christ carries the cross and is cruelly beaten by the Jews. 780 steps from the House of Pilate.'[34]

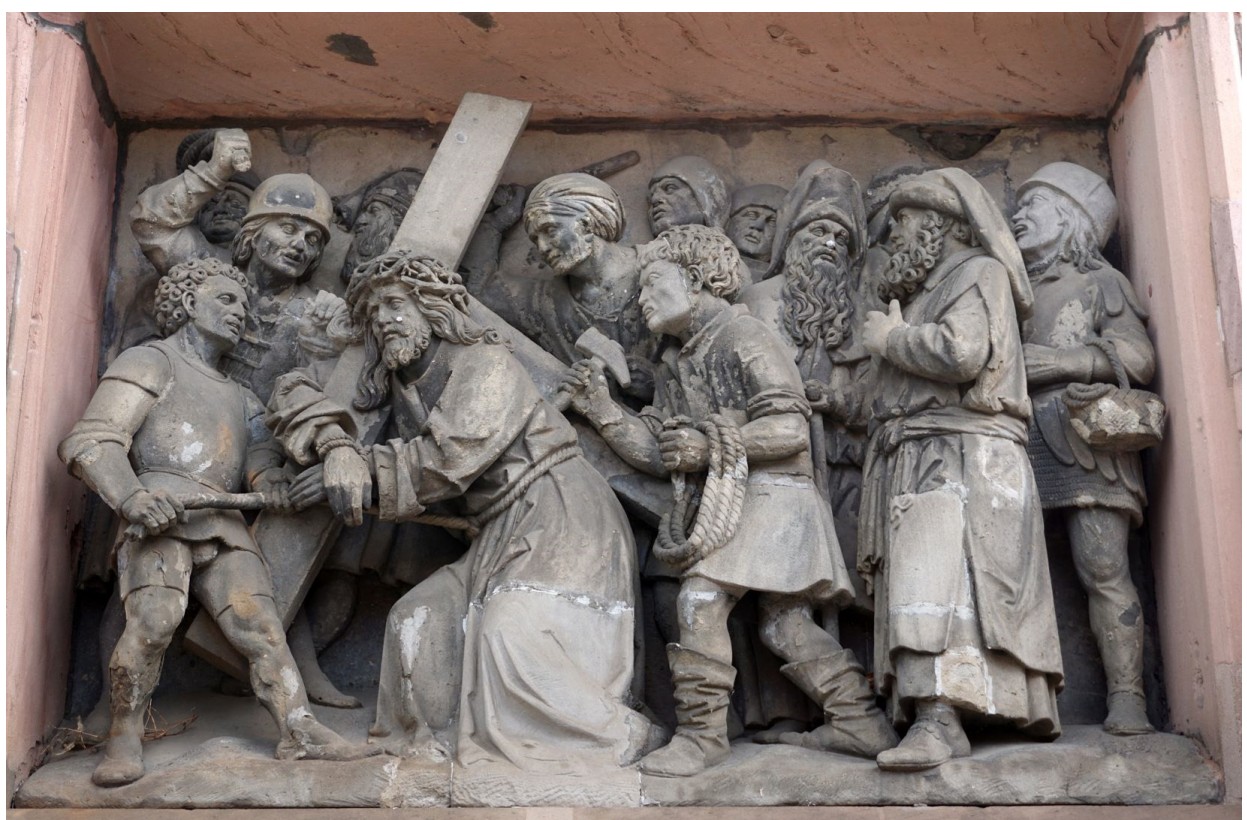

**Figure 19.** Kraft, *Stations of the Cross*, Nuremberg, Station Five (reconstruction).

Most dramatic of all the reliefs, number six (Figure 20) shows Jesus prostrate beneath the heavy cross, which extends diagonally across almost the entire width of the image. Directly above him, two tormentors in projecting high relief (considerably restored) bend over and reach across to pull both his arm and his hair, probably to force him upright again. The foot of the leftmost figure extends outward to touch the bottom of the frame. Behind them, a single second row of standing figures provides a well-spaced but varied backdrop of witnesses with a range of facial expressions from outright sneering to silent sympathy. Two men at the far right side converse; the bearded one wears a costly, fur-lined hat, and his beardless interlocutor is young, beardless, and lively, standing with clenched fists and a facial grimace. Here, Kraft fully displays his virtuoso command of modeling and relief carving, which was demonstrated previously in the Scheyer–Landau epitaph by the two character heads of Nicodemus and Joseph of Arimathea. A laconic inscription here is largely descriptive, and it omits the pathos of the relief itself: 'Here Christ falls to the ground from great weakness. 1100 steps from the House of Pilate.'[35]

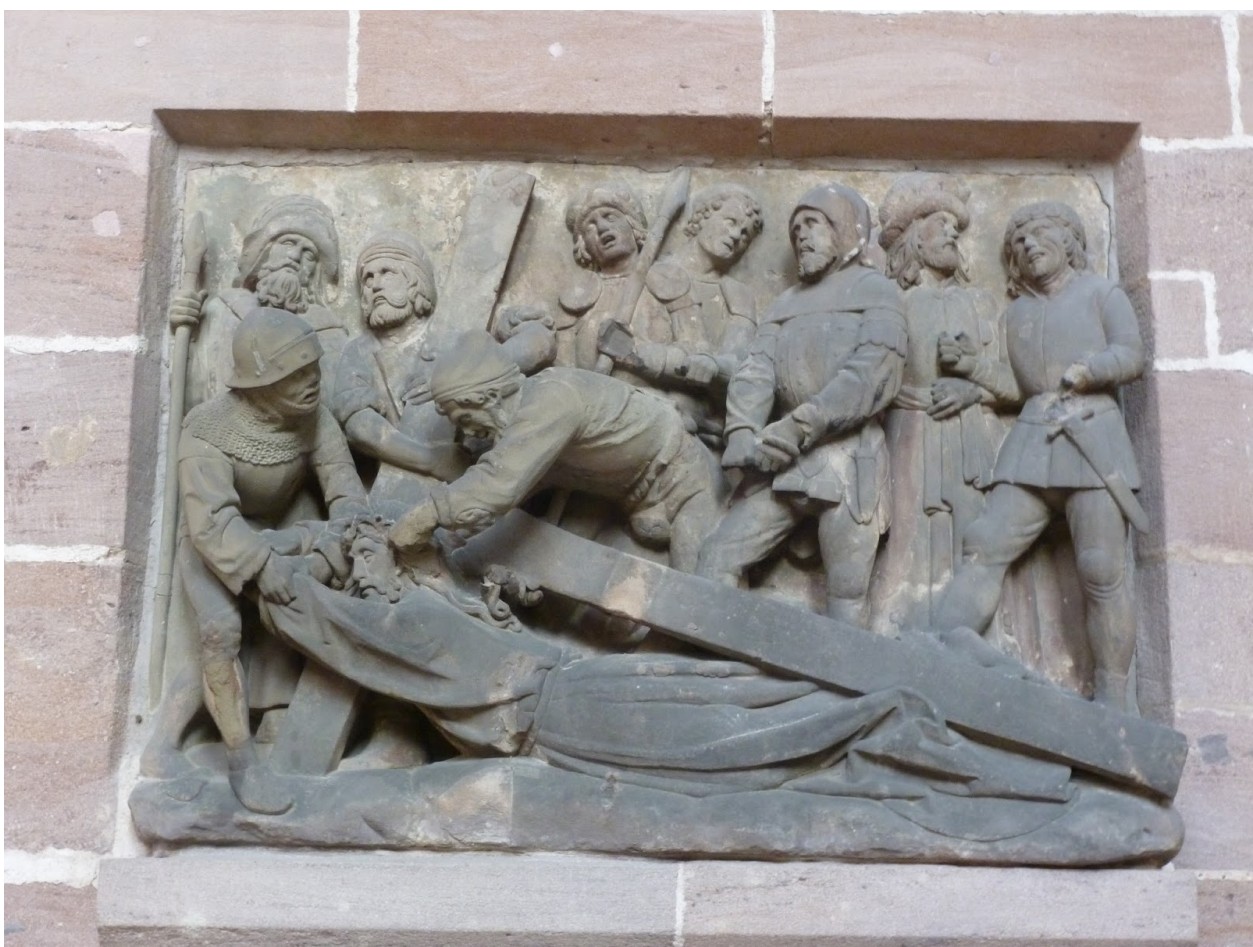

**Figure 20.** Kraft, *Stations of the Cross*, Nuremberg, Station Six (Nuremberg, Germanisches National-museum).

Kraft's relief sequence ends with the traditional *Lamentation* or *Deposition* (Figure 21), but it must have been placed to follow directly after the life-sized *Crucifixion* group, fully in the round, positioned at the entrance to St. John's Cemetery. In this final, seventh relief image, the dead body of Jesus is stretched into a seated L-shape at the lower left of the composition. This lone relief shows the exposed body of Jesus, and it demonstrates Kraft's command of anatomy and its representation, which is also seen nearby in the full-bodied figure in the round of Jesus on the cross (Figure 10).[36] Youthful St. John supports the corpse from behind, as the Virgin, clasping his jaw, tenderly kisses her son. Two other female mourners tenderly examine the wounds of his hand; the one with long, braided hair at Christ's feet is presumably Mary Magdalene. The oil ointment, her traditional attribute, is held above her head by a man in profile, who wears a conical hat, normally the distinctive marker of a Jew.[37] With her other hand, she wipes away tears, another frequent action of the Magdalene beneath the cross.

The entire seventh scene offers varied expressions of mourning, but the standing background row of figures appears more meditative and introspective about the event, providing another possible model for a contemplative pious beholder of the scene. Two women at upper left face forward with mournful faces and hands joined above their hearts, while a third turns to converse with a man beside her. Dressed expensively, he is surely wealthy Joseph of Arimathea, who paid for Christ's burial. He holds the Crown of Thorns, while a man beside him carries the four nails of martyrdom (the same roles as the two figures of the Scheyer–Landauer reliefs; cf. Figure 7). At the upper right, the man in profile with the distinctly Jewish hat, who holds the ointment jar used for burials, would be Nicodemus,

identified as a Pharisee. This final inscription underscores the pathos of the scene: 'Here Christ lies dead before his generous, worthy Mother, who mourns him with great heartache and bitter pain.'[38]

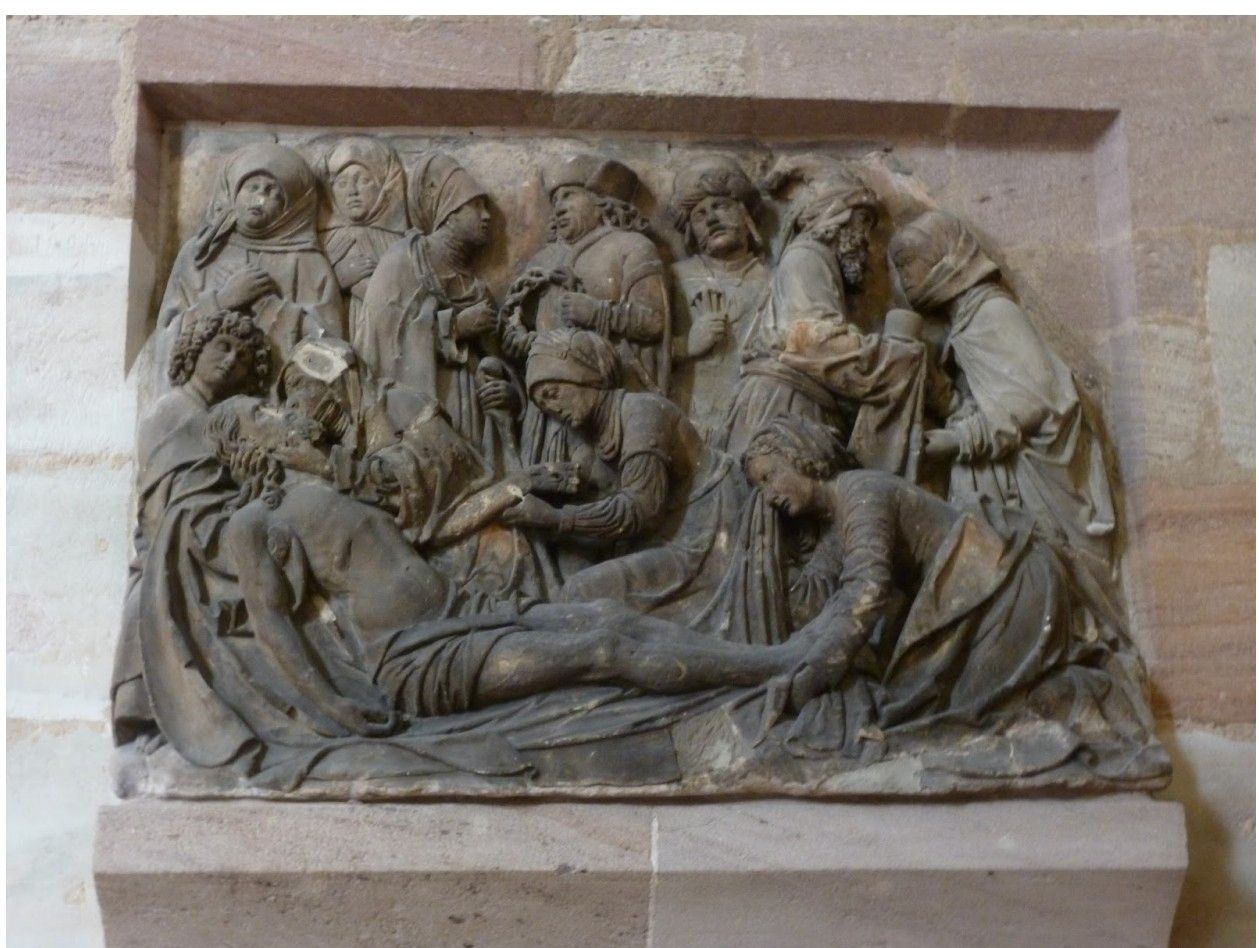

**Figure 21.** Kraft, *Stations of the Cross*, Nuremberg. Station Seven (Germanisches Nationalmuseum).

As noted, the Kraft reliefs find their culmination in the full-scale, three-dimensional figures of the *Crucifixion* group with three crosses and standing figures below, now preserved only in fragments. The most notable surviving figure is the frontal, balanced *Christ on the Cross*, today in Nuremberg's Heilig-Geist-Spital (Figure 10). This Jesus has flowing locks as well as an enlivened loincloth with deep folds, reminiscent, albeit more subdued, of the work of Veit Stoss in Nuremberg, especially his own contemporary work in limewood for the Heilig-Geist-Spital (c. 1505-10).[39]

An engraving by Johann Alexander Boener from around 1700 (Figure 22) shows the Crucifixion figure group beside the entry arch to the Cemetery and the large seventh relief, the *Lamentation*, on the opposite side of the gate, mounted on the enclosing wall and thus capable of being read in tandem with the *Crucifixion* group as its denouement.[40] The same print also shows the Holzschuher family burial chapel, dated 1508, nearby and enclosed to the left within the Cemetery. It appears as a rotunda with a prominent apse. Already in 1515 that building received an indulgence in its own right as a replica of the Holy Sepulcher, and its interior decoration, placed under a rounded arch, is that a multi-figure sculpted ensemble by Kraft of the *Entombment* (Figure 11), complemented with an anonymous painted background that suggests an imagined bird's-eye topographical view of Jerusalem.[41] Thus, it forms the ultimate destination as an event and a building to honor it as the Holy Sepulcher itself, and it culminates the Via Dolorosa in Jerusalem. Now

replicated in Nuremberg, the Holzschuher Chapel provides the final destination for the pious, local, virtual pilgrim. The same figures and costumes appear as in the *Lamentation* relief (Figure 21; Dürer's woodcuts of the same subjects provide the same continuity). Added to the sides of this sculpted chapel scene are sleeping Roman soldiers, proleptically suggesting the following event of the Resurrection from this same tomb.

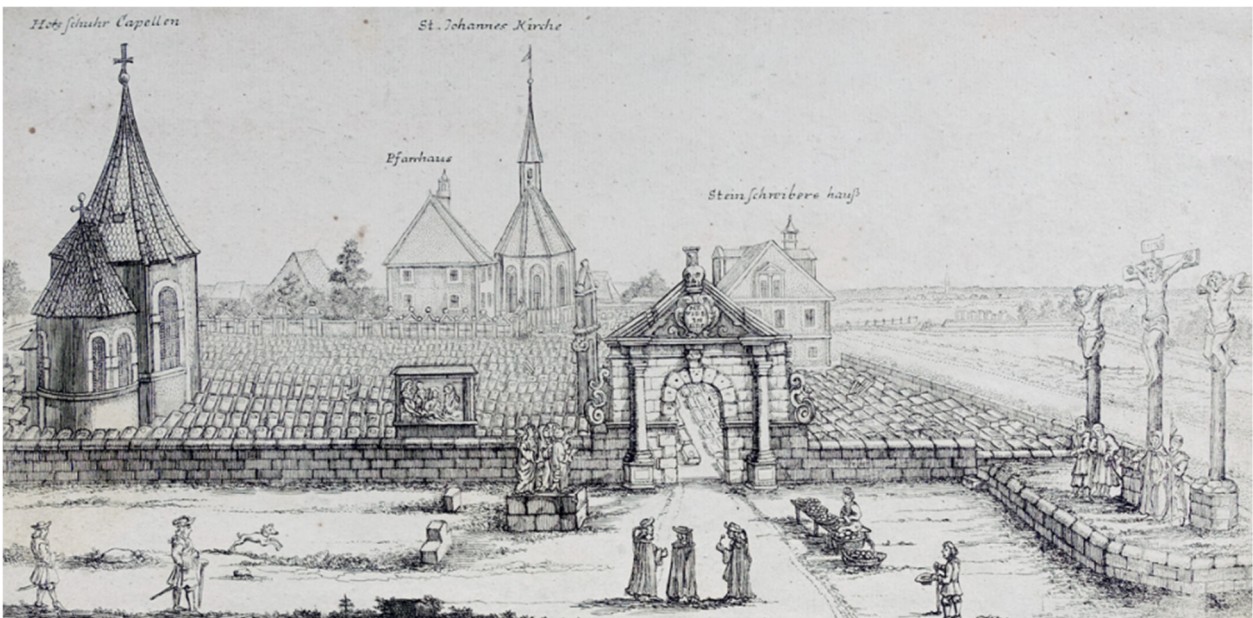

**Figure 22.** Johann Alexander Boener, *St. John's Cemetery, Nuremberg*, engraving, ca. 1700.

### 3. Imitatio Pietatis

Regardless of the order in which these two major relief projects by Kraft were made, both works provide a pious viewer with what can only be described as *moving* images. Both German and English languages use the same phrase 'to move' in both a physical and an emotional sense.

Physically, an observer certainly must actually move along both of Kraft's reliefs in sequence. Even at the U-shaped Schreyer–Landauer monument, the chronological sequence begins with *Christ Carrying the Cross* at the right side and then progresses right-to-left across a wide central vista to end at the Resurrection at left. In the *Stations of the Cross*, the viewer moves a considerable number of explicit steps westward from the city gates toward the St John's Cemetery, supposedly following the literal, historical footsteps of Jesus himself from the House of Pilate to Golgotha, the site of the Crucifixion and then onward to the interment of Christ's body in the Entombment (now located within the contemporary walls of the local cemetery itself). In Frank Matthias Kammel's felicitous phrase, the exact number of those footsteps is essential: "The recounting is closely bound up with the counting."[42] Because each individual relief of the *Stations of the Cross* moves visually from right to left, it conveys both the weight and struggle of bearing the cross, especially against the inflicted pains from tormentors along the Via Dolorosa. Also noteworthy is that Kraft's horizontal format of the Passion was taken up by Dürer in his latest version of the multi-figure sequence, his so-called "Oblong Passion" drawings of the early 1520s (Figure 23).[43]

But the other sense of the phrase 'to move' is mental: to bestir the emotions, in this case the complex viewer empathy for the sufferings of Jesus, whose own mental stress, already called *Angst* in German art during his anguished overnight foreknowledge of the coming Passion and Crucifixion while he meditated on the Mount of Olives.[44] Dürer's own com-Passion extended to a literal identification: a self-portrait drawing as Christ, the *Man of Sorrows* (1522; W. 886; formerly Bremen; Figure 24), holding his whips and scourges.[45]

He also empathized the individual *imitatio Christi* seriously enough to produce another pair of pendant drawings (c. 1523; London, British Museum) with Jesus bearing the cross on the heraldically favorable left half, and a prayerful Christian in profile with his hands crossed in prayer, who symbolically bears his own cross on the right half.[46]

As these comparisons to Dürer attest, Adam Kraft, his Nuremberg contemporary, worked in the material of sandstone to provide a counterpoint experience in sandstone relief to represent the Passion of Christ. Both artists began their work in Nuremberg around the same time, 1490, although the older Kraft predeceased Dürer by two full decades (1508/1528). But both Nuremberg artists shared that religious sentiment of late-medieval art to evoke pious emotions through vivid, multi-figured narrative re-enactments. In Kraft's *Stations of the Cross* (Figure 25), the artist simulated a virtual surrogate pilgrimage, as if re-enacted in Jerusalem itself. Through their visual process, both Kraft and Dürer moved pious empathy in their—literally—moving Passion sequences.

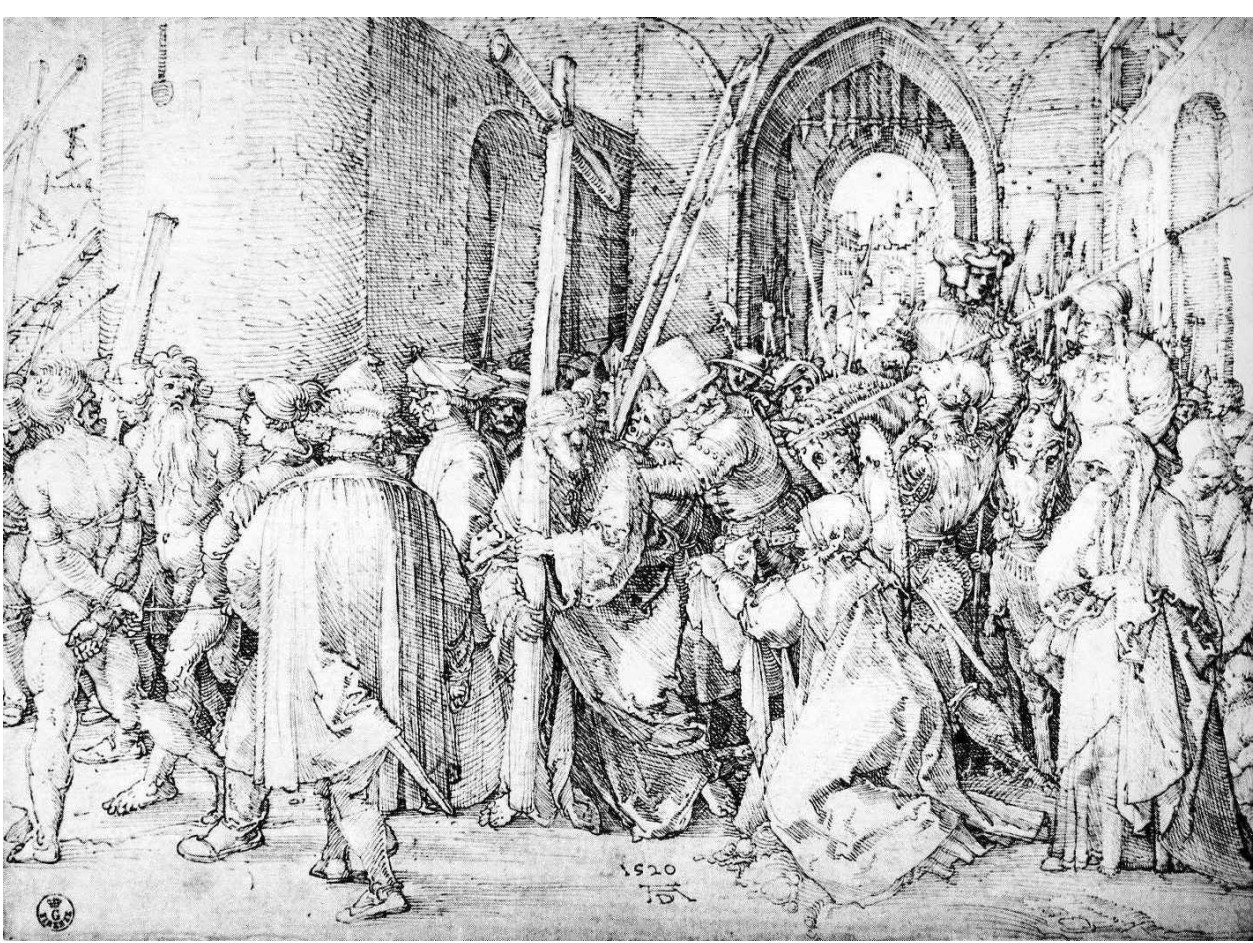

**Figure 23.** Albrecht Dürer, *Christ Carrying the Cross*, ink drawing, 1520 (Florence, Uffizi Gallery).

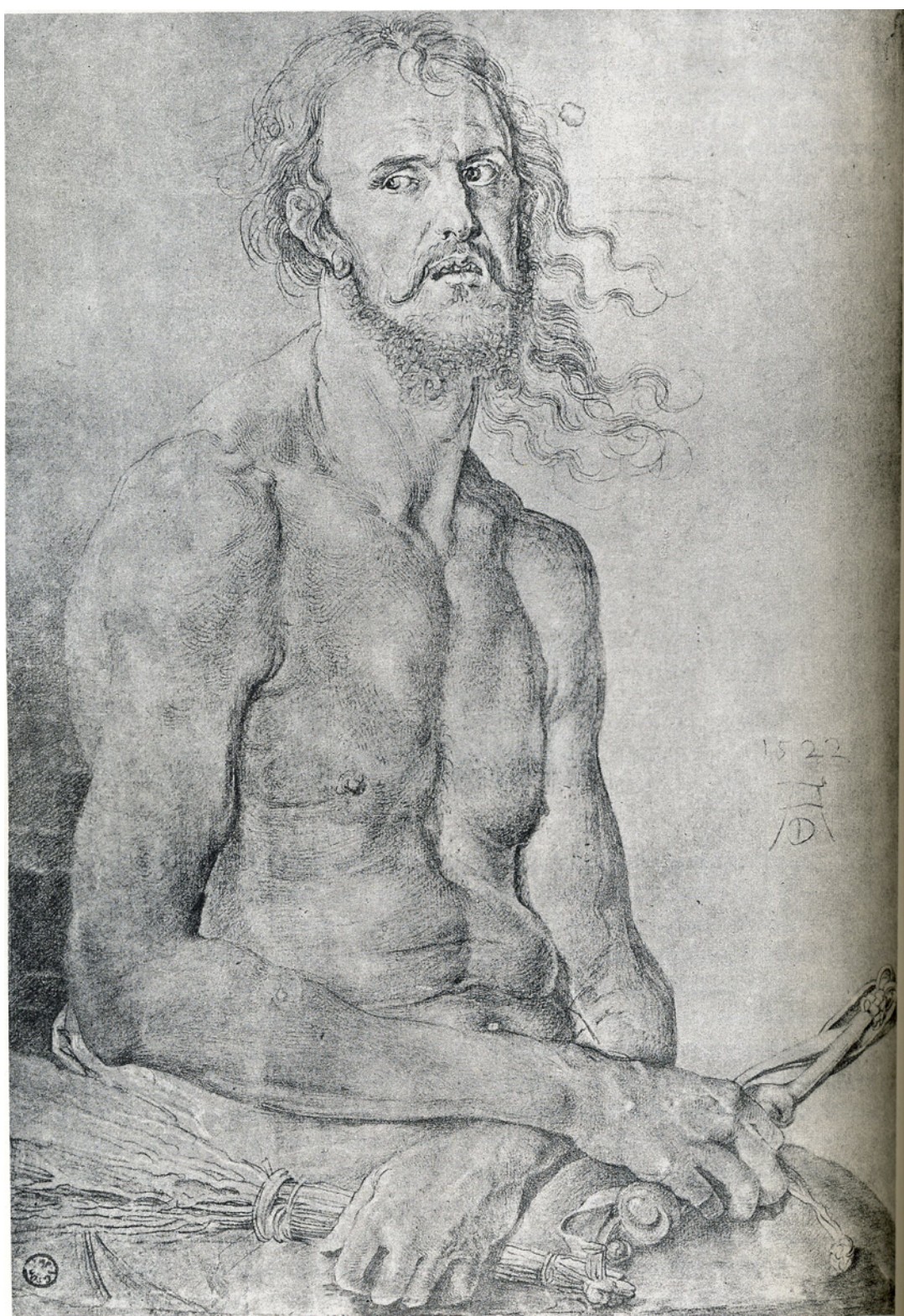

**Figure 24.** Albrecht Dürer, *Self-Portrait as Man of Sorrows*, ink drawing, 1522 (formerly Bremen, Kunsthalle, lost).

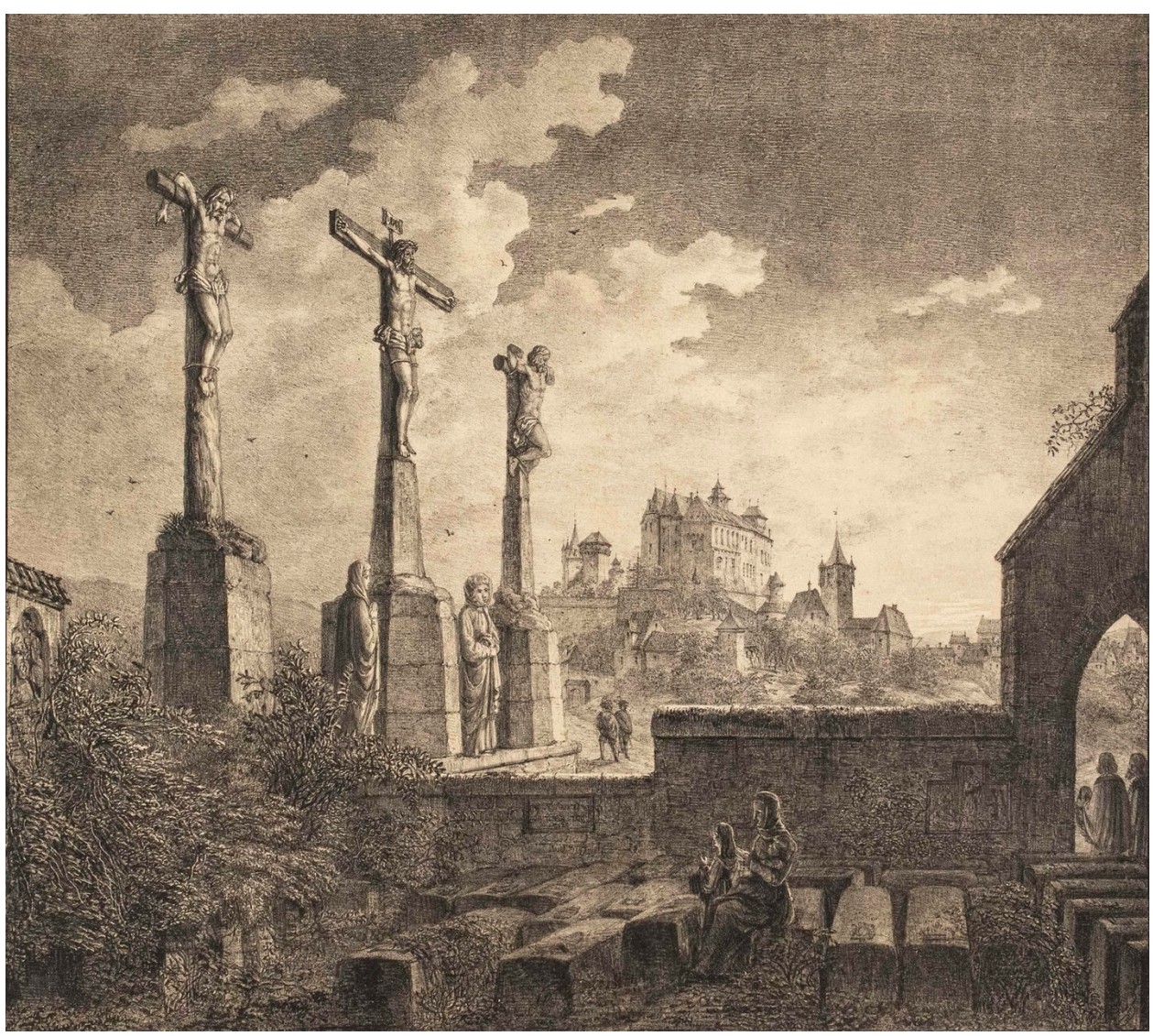

**Figure 25.** Domenico Quaglio II (German, 1787–1837), The Graveyard of St. John's Church with a View of Nuremberg Castle, lithograph, 1819.

**Funding:** This research received no external funding.

**Conflicts of Interest:** The author declare no conflict of interest.

## Notes

1    Kahsnitz, *Stoss in Nürnberg*, pp. 218–58, no. 20.

2    Corine Schleif, *Donatio et Memoria. Stifter, Stiftungen und Motivationen an Beispielen aus der Lorenzkirch in Nürnberg* (Munich, 1990), pp. 18–45; Schleif, '500 Jahre Sakramenthaus: Erklärung-Verklärung, Deutung-Umdeutung. *St.Lorenz 96. Mitteilung des Vereins zur Erhaltung der Lorenzkirche* NF 41 (1996), pp. 3–47.

3    On the former Corine Schleif, 'Nicodemus and Sculptors: Self-Reflexivity in Works by Adam Kraft and Tilman Riemenschneider,' *Art Bulletin* 75 (1993), pp. 599–603; *St. Sebald. 500 Jahre Grabmal der Familien Schrey und Landauer von Adam Kraft* (Nuremberg, 2000). For the latter, Frank Matthias Kammel, ed., *Adam Kraft. Der Kreuzweg*, exh. cat. (Nuremberg: Germanisches Nationalmuseum, 2018), is fundamental.

4    Some current scholarship suggests that the Passion reliefs might predate the St. Sebald's epitaph and thus stem from Kraft's earliest Nuremberg activity, ca. 1490; however, the presence of a complementary full-figured *Entombment* by Kraft at the Cemetery site in the Holzschuher Chapel, dated 1508, plus the rendering of a full-scale, three-dimensional *Crucifixion* ensemble (preserved only in fragments) at the same site, reveals a mature sculptor with a large workshop, well accustomed to working in the round.

Since that added element seems directly tied to the relief sequence of the *Stations of the Cross*, it suggests that the more traditional dating from the end of Kraft's career, ca. 1506-08, could be more reasonable. Kammel, *Kraft Kreuzweg*, pp. 47–48; Reiner Zittlau, *Heiliggrabkapelle und Kreuzweg. Eine Bauaufgabe in Nürnberg um 1500* (Nuremberg, 1992), pp. 70–92.

[5] Schleif, 'Nicodemus and Sculptors,' pp. 599–601. Their request was submitted to the same Paul Volckamer, trustee for the church, who would commission Stoss in the following year for his own epitaph inside St. Sebald's. The church officials then took the request to the city council, further demonstrating their central control over Nuremberg's civic monuments.

[6] Paul Schoenen, 'Epitaph,' *Reallexikon zur deutschen Kunstgeschichte* V (Stuttgart, 1967), cols. 872–921, esp. col. 873. See the related Netherlandish tradition: Douglas Brine, 'Jan van Eyck, Canon Joris an der Paele, and the Art of Commemoration,' *Art Bulletin* 96 (2014), pp. 265–87.

[7] Fedja Anzelewsky, *Albrecht Dürer. Das malerische Werk* (Berlin, 1971), pp. 159–60, 173–74, nos. 55, 70. Compare to Dürer's *Paumgartner Altarpiece* (1498/1503; Munich, Alte Pinakothek), where similarly small donor figures appear beneath a central *Nativity* ibid., 156, no. 50.

[8] Rubens would feature a heroic, resurrected Christ on several epitaphs; David Freedberg, 'Rubens as a Painter of Epitaphs, 1612–1618' *Gentsche Bijdragen tot de Kunstgeschiedenis* 24 (1976), pp. 51–71.

[9] Joseph of Arimathea was the wealthy man who took the bod of Jesus for burial (John 19: 38–39; Matthew 27: 57–58). Veneration of the instruments of the Passion as the *arma Christi* formed a prominent late medieval cult; Gertrud Schiller, *Iconography of Christian Art* (London, 1972), vol. II, pp. 189–96. For the self-portrait, Schleif, 'Nicodemus and Sculptors,' pp. 599–603, compares the features of the bearded Nicodemus to the celebrated kneeling Kraft self-portrait underneath the St. Lorenz tabernacle. His companion has been identified as Sebald Shreyer, but his beard makes that portrait comparison unlikely. However, the other donor, Matthias Landauer, did have a prominent beard later, as shown in his profile donor portrait in Dürer's 1511 *Adoration of the Trinity* (Vienna, Kunsthistorisches Museum; Anzelewsky, *Dürer malerische Werk*, fig. 145). Moreover, Landauer's grave was located directly below that same image of the three crosses. Dürer would later insert his own portrait as a witness in several paintings, even alongside his friend Conrad Celtis in the *Martyrdom of the 10,000* (1508; Vienna, Kh. Museum; Schleif, ibid., p. 623, figs. 32–33).

[10] Andrew Robison and Klaus Albrecht Schröder, eds., *Albrecht Dürer. Master Drawings, Watercolors, and Prints from the Albertina*, exh. cat. (Washington: National Gallery, 2013), pp. 148–56, nos. 43–47; Schröder and Maria Luise Sternath, eds., *Albrecht Dürer*, exh. cat. (Vienna: Albertina, 2003), pp. 324-24, nos. 89–96. See also his unfinished late sequence of Passion drawings: Dana Cowen, 'Albrecht Dürer's Late Passion Drawing: The *Oblong Passion* in Context,' in Susan Foister and Peter van den Brink, eds., *Dürer's Journeys*, exh. cat. (London: National Gallery, 2021), pp. 241–51.

[11] Jordan Kanter, *Dürer's Passions* (Cambridge, MA, 2000); for Schongauer, Charles Minott, *Martin Schongauer* (New York, 1971), pp. 42–43; nos. 19–28, plus additional engravings of the *Harrowing of Hell* (no. 29) and *Resurrection* (no. 30).

[12] Again, the precise date of the *Kreuzweg* remains uncertain. For the city locations, see the 1608 Hieronymus Braun city map in Daniel Hess and Thomas Eser, eds., *The Early Dürer*, exh. cat. (Nuremberg: Germanisches Nationalmuseum, 2012), pp. 598–603, where the Dürerhaus is no. 40 and St. Sebald's is no. 58.

[13] F.O. Büttner, *Imitatio pietatis. Motive der christlichen Ikonographie als Modelle zur Verähnli-chung* (Berlin, 1983), esp. pp. 56–62 for Christ Carrying the Cross; James Marrow, 'Inventing the Passion in the Late Middle Ages,' in Marcia Kupfer, ed., *The Passion Story. From Visual Repre-sentation to Social Drama* (University Park, PA, 2008), pp. 23–52; Marrow, *Passion Iconography in Northern European Art of the Late Middle Ages and Early Renaissance* (Kortrijk, 1979).

[14] David Hotchkiss Price, *Albrecht Dürer's Renaissance* (Ann Arbor, 2003), pp. 137–38, 169–93, noting, p. 171, that in the late fifteenth century Nuremberg's government pursued the goal of banishing its Jewish residents. Numerous images of historical Jewish persecutions also appear among the woodcuts of Hartmann Schedel's 1493 Nuremberg *World Chronicle*, including the infamous martyrdom of Simon of Trent (1475; fo. 254v), blamed on that local Jewish community.

[15] Price, *Dürer's Renaissance*, p. 181. The verses accompanying the violent *Crowning with Thorns* from the *Small Woodcut Passion* is still more emphatic: 'It is not enough that they [the Jews] cut Christ to ribbons with their bramble-whips . . . he is spit upon, bashed, drubbed with cudgels, ripped from his lofty throne and dragged by his hair.' Price, ibid., p. 186, itemizes the numerous *Small Passion* woodcuts that emphasize torture of Christ's perfect body by Jews: *Christ before Annas; Christ before Caiaphas; Christ Mocked; Christ before Pilate; Christ Scourged; Christ Crowned with Thorns; Ecce Homo; and Christ Nailed to the Cross.*

[16] Kammel, *Kraft Kreuzweg*, pp. 48–49. Neudörffer's account is 1546. The patronage of Ketzel was assigned by local jurist Christoph Friedrich Gugel (1682); however, 1905 research by St Sebald parson (*Pfarrer*) Christian Geyer revised several such accumulated legends about the Kraft Stations of the Cross. Martin Ketzel's pilgrimage to the Holy Land also left visual traces in family records (Kammel, ibid., figs. 26–27). For the 1500/03 Bamberg Via Dolorosa sponsored by Marschalk, Kammel, ibid., pp. 53–55, figs. 40–41.

[17] Kammel, *Kraft Kreuzweg*, pp, 18–19; Christopher Wood, *Forgery Replica Fiction. Temporalities of German Renaissance Art* (Chicago, 2008), pp. 47–53

[18] Kammel, *Kraft Kreuzweg*, pp. 22–31; Sarah Lenzi, *The Stations of the Cross. The Placelessness of Medieval Christian Piety* (Turnhout, 2016). Walter Haug and Burghart Wachinger, eds., *Die Passion Christi in Literatur und Kunst des Spätmittelalters* (Tübingen, 1993),

esp. essays by Fritz Oskar Schuppiser, ibid., pp. 169–210; and Jörg Fichte, ibid., 277–96. For French Passion play mss. and images, Laura Weigert, *French Visual Culture and the Making of Medieval Theater* (Cambridge, 2015), pp. 74–124.

[19] Carol Schuler, 'The Seven Sorrows of the Virgin: Popular Culture and Cultic Imagery in Pre-Reformation Europe,' *Simiolus* 21 (1992), pp. 5–28. On the *arma Christi*, Rudolf Berliner, 'Arma Christi,' *Münchner Jahrbuch der Bildenden Kunst* 6 (1955), pp. 35–152; Robert Suckale, 'Arma Christi: Überlegungen zur Zeichenhaftigkeit mittelalterlicher Andachtsbilder,' *Städel-Jahrbuch* 6 (1977), pp. 177–207. Connecting Passion images, esp. the Man of Sorrows, to memory images, Peter Parshall, 'The Art of Memory and the Passion,' *Art Bulletin* 81 (1999), pp. 456–72. Also for pictorial sanctification, Thomas Lentes, '"As far as the eye can see...": Rituals of Gazing in the Late Middle Ages," in Jeffrey Hamburger and Anne-Marie Bouché, eds., *The Mind's Eye. Art and Theological Argument in the Middle Ages* (Princeton, 2006), pp. 360–73.

[20] Kathryn Rudy, *Virtual Pilgrimages in the Convent: Imagining Jerusalem in the Late Middle Ages* (Turnhout, 2011), esp. pp. 58–90; for the relation of sites to the images in a major printed pilgrimage guide, Bernhard von Breidenbach's *Peregrinatio in terram sanctam* (Mainz, 1486), Elizabeth Ross, *Picturing Experience in the Early Printed Book. Breydenbach's* **Peregrinatio** *from Venice to Jerusalem* (University Park, PA, 2014), esp. pp. 157–64.

[21] *The Itineraries of William Wey* (London, 1857; trans. Oxford, 2010). Nuremberg pilgrim Hans Tucher the Elder took a tour led by Franciscans in 1479 and even counted the steps, a significant, specific fact recorded on the Kraft inscriptions (see below). Kammel, *Kraft Kreuzweg*, p. 23, notes other prominent German pilgrims, such as Wittelsbach elector Ottheinrich von der Pfalz in 1521.

[22] Dirk De Vos, *Hans Memling* (Ghent, 1994), pp. 105–9, no. 11. A lone surviving two-sheet woodcut by Swiss artist Urs Graf (London, British Museum) also shows various Passion scenes in a hilly landscape but also includes two pilgrim visitors in the lower center, suggesting that it might in fact be a "Passion park" like the Sacro Monte at Varallo; Rudy, *Virtual Pilgrimages*, pp. 248–51, figs. 91–92. For Varallo, David Freedberg, *The Power of Images* (Chicago, 1989), pp. 192–200.

[23] Kenneth Nebenzahl, *Maps of the Holy Land* (New York, 1986), pp. 90–91. The final work was published posthumously as *Theatrum terrae sanctae* in Cologne in 1590.

[24] For earlier reliefs in Germany, including a 14th-century set in the east choir of St. Sebald's itself, Kammel, *Kraft Kreuzweg*, pp. 25–28, figs. 11–15, and for earlier *Stations* emerging from city walls in Germany, ibid., pp. 31–36, figs. 18–23.

[25] For the Lentulus letter, Lloyd DeWitt, 'Testing Tradition against Nature: Rembrandt's Radical New Image of Jesus,' in DeWitt, ed., *Rembrandt and the Face of Jesus*, exh. cat. (Paris-Philadelphia-Detroit, 2011), pp. 109–45, esp. pp. 112–23. For the side wound, David Areford, 'The Passion Measured: A Late-Medieval Diagram of the Body of Christ,' in A.A. McDonald and Bernhard Ridderbos, eds., *The Broken Body: Passion Devotion in Late Medieval Culture* (Groningen, 1998), pp. 211–38.

[26] Kammel, *Kraft Kreuzweg*, pp. 37–40. In the Netherlands the painter Jan van Scorel made his own pilgrimage to Jerusalem in 1520. Thus, he included its topography as an accurate background in his *Entry of Jesus into Jerusalem* on the Lokhorst triptych (c. 1526; Utrecht, Centraal Museum). He also included his self-portrait as a fellow member within one of the first Netherlandish independent group portraits, the *Jerusalem Brotherhood at Haarlem* (c. 1528; Haarlem, Frans Halsmuseum), which shows all of these former pilgrims carrying palms like Jesus and facing a panel image of the Holy Sepulchre.

[27] Robert Suckale, *Die Erneuerung der Malkunst vor Dürer* (Petersberg, 2009), pp. 81–87, figs. 116–123.

[28] This torture is mentioned in some Passion narratives and appears in period images of the Carrying of the Cross; Marrow, *Passion Iconography*, pp. 171–89. For example, it also appears on two roughly contemporary paintings by Jheronimus Bosch of *Christ Carrying the Cross* (Vienna, Kunsthistorisches Museum; Escorial); Matthijs Ilsink et al., *Hieronymus Bosch. Painter and Draughtsman. Catalogue Raisonné* (Brussels, 2016), pp. 236-59, nos. 12–13.

[29] Quoted in Kammel, *Kraft Kreuzweg*, p. 43: *Hi[e]r begegnet Cristus seiner wirdigen lieben Mut[t]er die vor grossem herzenleit amechtig war IIc Srytt von Pilatus haus.*

[30] Kammel, *Kraft Kreuzweg*, p. 45: *Hie[r] ward Symon gezwungen Cristo sein krewtz helfen tragen IIcLXXXXV Sryt von Pilatus haus.*

[31] Büttner, *Imitatio pietatis.*

[32] Kammel, *Kraft Kreuzweg*, p. 45: *Hi[e]r sprach Cristus i[h]r Döchter von Jherulsale[m] ni[ch]t weynt vber mich sunder vber euch un[d] ewre kinder IIIcLXXX Srytt von Pilatus haws.*

[33] Kammel, *Kraft Kreuzweg*, p. 45: *hier hat Cristus sein heiligs angesicht der heiligen Fraw Veronika auf iren Slayr gedruckt vor irem Haws. Vc Stryt von Pilatus Haws.*

[34] Kammel, *Kraft Kreuzweg*, pp. 45–46: *Hier tregt Chrisuts das Crewtz vnd wird von den Juden se[h]r hart geslagen VIIcLXXX Srytt von Pilatus Haus.*

[35] Kammel, *Kraft Kreuzweg*, p. 46: *Hi[e]r fel]l]t Cristus vor grosser anmacht auf die Erden bey Mc Srytt von Pilatus haws.*

[36] Corine Schleif, 'Christ Bared: Problems of Viewing and Powers of Exposing,' in Sherry Lindquist, ed., *Meanings of Nudity in Medieval Art* (Farnham, 2011), pp. 251–78, esp. 266–73 on the woodcut sequence of the Seven Falls of Jesus. Kammel, *Kraft Kreuzweg*, p. 48, points to a Nuremberg precedent by Michael Wolgemut, the epitaph *Lamentation* (c. 1484; Nuremberg, St. Lorenz); see *Michael Wolgemut. Mehr als Dürers Lehrer*, exh. cat. (Nuremberg, 2019), pp. 211–3, no. 32. Of course, Dürer's own *Lamentation* from his Large Woodcut Passion (c. 1498/99) also provided an earlier model. Rainer Schoch, Matthias Mende, and Anna Scherbaum, *Dürer. Das druckgraphische Werk. II. Holzschnitte und Holzschnittfolge* (Munich, 2002), pp. 202–4, no. 162.

[37] Ruth Mellnkoff, *Outcasts: Signs of Otherness in Northern European Art of the Late Middle Ages* (Berkeley, 1993), esp. pp. 63–76, 91–94.

38  Kammel, *Kraft Kreuzweg*, p. 47: *Hi[e]r leyt Cristus tot vor seiner gebendeyten wirdigen Mut[t]er die i[h]n mit grossem Hertzenleyt vnd bitterlichen smertz claget vnd beweynt.*

39  Kahsnitz, *Stoss in Nürnberg*, pp. 122–28, no. 5; an even more flamboyant *Crucifix* by Stoss for St. Lorenz dates from a decade later, 1516–20; ibid., pp. 186–94, no. 16. The Stoss influence is perhaps another reason to incline towards a later dating of the Kraft *Crucifixion* group, at least, and Kammel concurs, *Kraft Kreuzweg*, p. 49, fig. 36.

40  Kammel, *Kraft Kreuzweg*, pp. 50–53, figs. 37–39.

41  Wood, *Forgery Replica Fiction*, pp. 47–53 for replicas of the Holy Sepulchre, such as Görlitz. Similar life-sized *Entombment* groups also appeared in France; William Forsyth, *The Entombment of Christ. French Sculptures of the Fifteenth and Sixteenth Centuries* (Cambridge, MA, 1970).

42  Kammel, *Kraft Kreuzweg*, p. 59: ' . . . *ist das Erzählen eng verbunden mit dem Zählen.*'

43  Dana Cowen, '*Oblong Passion*,' pp. 241–51.

44  Büttner, *Imitatio Pietatis*, pp. 47–55; Larry Silver, 'The Influence of Anxiety: The Agony in the Garden as Artistic Theme in the Era of Dürer,' *Umeni* 45(1997), pp. 420–9; Donald McColl, 'Agony in the Garden: Dürer's "Crisis of the Image,"' in Larry Silver and Jeffrey Chipps Smith, eds., *The Essential Dürer* (Philadelphia, 2010), pp. 166–84.

45  McColl, 'Agony in the Garden,' pp. 175–6, fig. 10.5; Joseph Koerner, *The Moment of Self-Portraiture in German Renaissance Art* (Chicago, 1993), p. 179, fig. 96.

46  Büttner, *Imitatio Pietatis*, pp. 56–62, figs. 50–51; McColl, 'Agony in the Garden,' pp. 175–7, fig. 10.6; Koerner, *Moment of Self-Portraiture*, pp. 76–77, fig. 35

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
