# Peer review of "Adam Kraft’s Moving Sandstones"

_arts_

Round 1
Reviewer 1 Report (New Reviewer)
This is a great topic, well worthy of writing about in English. Your approach is also timely. I think, though, that you might be trying to cram too many ideas into one article.

Author Response
The boxes and their evaluations are too general to be addressed individually and give a truly distorted view of language issues, which led to a call to have a 'native speaker' (I AM a native speaker) assist with the editing process. I am working through the text line by line to improve clarity. As for the abundance of ideas, I see that as a virtue rather than a liability--addressing the topical issue of affective piety for the volume as well with attention to period goals of the art as well the Nuremberg specifics of location and material.
Reviewer 2 Report (New Reviewer)
A key component of the argument hinges on the right-to-left reading of both the Schreyer-Landauer Epitaph and the Kreuzweg sculptures. The author asserts that the right-to-left orientation was a mechanism by which Kraft tried to cause discomfort in the viewer, in order to mirror the discomfort Christ himself experienced in the Passion. I wonder, however, if there may have been pragmatic reasons behind Kraft's decisions. For example, the location of the epitaph on the exterior of the Sebalduskirche apse reads "correctly" for those walking past it from the Kaiserburg down to the Hauptmarkt, a processional route used for city festivities such as the Schembartlauf. (Was this also the direction that the shrine of St. Sebaldus traveled in the annual procession of the saint’s relics through the city?) Likewise, presumably the decision to locate the Kreuzweg sculptures on the right-hand side of the road leading to the Johannisfriedhof was established prior to Kraft’s beginning the project. Would that not account for the directionality of the characters depicted? Furthermore, the Bamberg Kreuzweg, which predates Kraft’s but was commissioned by the same individual, also reads from right-to-left.
The author also ties in Dürer’s print series depicting the Passion, which reads left-to-right, but those works postdate Kraft’s sculptures. Perhaps Schongauer’s engraved Passion (produced around 1480) would be a more fruitful comparison as it also reads left-to-right. However, his masterpiece single-sheet engraving of Christ Carrying the Cross of c. 1475-80 depicts Christ heading leftward towards Golgotha. Again, the issue of directionality as a main component of the author’s thesis becomes a bit muddled when additional comparisons are examined.
Line 117: Dürer did not move into the house on the Tiergärtnertor until 1509, although he would certainly have known of Kraft’s stations of the cross sculptures even without this eventual proximity to the starting location.
Paragraphs beginning line 118, regarding anti-Semitism in Passion imagery: Keep in mind that the Frauenkirche on the Hauptmarkt, the building that housed the German Imperial Regalia in a shrine hung from the choir ceiling, was built on the site of the Jewish Synagogue, destroyed in 1349 when the Jewish community of Nuremberg was exiled. The fact that Nicodemus (as a self-portrait) and Joseph of Arimathea (as a donor portrait) are also portrayed as Jews despite being sympathetic characters in the Passion story accords with late medieval devotional practice of aligning Christian sinners’ imperfect piety as equivalent to the violent actions carried out against Christ.
Image on page 15: highly recommend procuring a better, clearer image that does not have a digital date stamp
I am concerned that nowhere does the author reference Gerhard Weilandt’s excellent monograph on St. Sebald when discussing the Schreyer-Landauer Epitaph. (See Jacqueline Jung’s review of the book here: https://arthist.net/reviews/359/lang=en_US)
There are occasional typos and grammatical incoherencies in the manuscript. Another thorough proofread is warranted.
Author Response
This is a very helpful review with good specifics about Nuremberg streets that I need to consider, although the layout remains the same, whichever direction a viewer approaches the images. Therefore, my contention will still probably still stand concerning the right-to-left arrangement, which is unusual in relation to both the comparative prints that I cite (especially by Albrecht Dürer) as well as numerous conventional paintings and reliefs, all of which read from left to right. Of course, that point is not the major core of the argument, but might need some clarification within the wider presentation.
This manuscript is a resubmission of an earlier submission. The following is a list of the peer review reports and author responses from that submission.
Round 1
Reviewer 1 Report
The article lacks an abstract and does not explain the purpose of the study in the introductory paragraph. This paragraph establishes a historical context, but I do not see the relationship with the rest of the text. In fact, there is no introduction as such, specifying the objectives and establishing a State of the Question. Likewise, the article lacks conclusions. A bibliographic could also be added.
Thus, it is not clear what the article contributes to the knowledge of the work of Adam Kraft, beyond describing the images.
In line 33, the author says that "current scholarship suggests that the Passion reliefs predate the epitaph...", but does not specify exactly the references that support this hypothesis.
As for the formal aspects, the author has not met one of the requirements: "All Figures, Schemes and Tables should be inserted into the main text close to their first citation and must be numbered following their number of appearance (Figure 1, Scheme 1, Figures 2, Scheme II, Table 1, etc.)"
The text has a verb-less phrases like the one in line 24.
While on line 114, the comma behind the subject should be removed.
Some references lack data such as date: for example, in footnote 3 or footnote 37.
Author Response
I will make an effort to insert figures and cues to them in the paper and make an introductory paragraph to assert the hypothesis and argument of the paper. I will trim the introductory discussion to make the focus clearer.
This reader is clearly a non-specialist, however, because in insisting on images, s/he raises a problem that I did not encounter on the more useful responses the first time that I published in Arts. But I am remedying that omission in the revised paper.
The grammatical points are niggling and arbitrary and just questions of style.
Reviewer 2 Report
This is a largely descriptive account of Adam Kraft's Schreyer-Landau reliefs. Symptomatic of the lack of any overt, overarching critical thinking, it has neither an introduction nor a conclusion: the author nowhere states either the purpose of the article or what the reader might hope to learn from this presentation of the material. The author needs to ask him/herself what questions / issues s/he is trying to address and how they relate to existing scholarship: s/he must then state them clearly at the outset, rework the entire piece so that it addresses them, and add a conclusion explaining the findings both in relation to the cycle that is the particular focus, and for the role of such artworks in contemporary society more generally. Perhaps illustrations will be supplied in due course; however, none accompanied the version sent the present reader - a lack that compromises intelligibility even for someone who has seen the works. Some of the prose is non-standard or awkward and needs to be improved - though there is no point trying to remedy this until the fundamental issue of the absence of an argument / purpose to the enquiry has been addressed.
Author Response
As in my response to Reader One, I am taking pains to make the argument clearer at the outset. However, humanities papers are not as structured (one might even say stilted) as scientific papers, which often resemble lab reports.
Description is necessary for the images to bring out the very affective aspects of the narration, and now that the photos will be included, perhaps the later readers will see better what the point is about their emphasis on physical burdens and emotional torments. That is where the paper responds to the editor's call for an artwork (here two of them) that powerfully moves viewer emotions. I should also note, re the descriptive elements of the paper, that these works have not been well studied, esp in English, so they need to be presented to readers with some attention to detail.